# Learning diffusion at lightspeed

**Antonio Terpin**
ETH Zürich
aterpin@ethz.ch

**Nicolas Lanzetti**
ETH Zürich
lnicolas@ethz.ch

**Martín Gadea**
ETH Zürich
mgadea@ethz.ch

**Florian Dörfler**
ETH Zürich
dorfler@ethz.ch

## Abstract

Diffusion regulates numerous natural processes and the dynamics of many successful generative models. Existing models to learn the diffusion terms from observational data rely on complex bilevel optimization problems and model only the drift of the system. We propose a new simple model, JKOnet*, which bypasses the complexity of existing architectures while presenting significantly enhanced representational capabilities: JKOnet* recovers the potential, interaction, and internal energy components of the underlying diffusion process. JKOnet* minimizes a simple quadratic loss and outperforms other baselines in terms of sample efficiency, computational complexity, and accuracy. Additionally, JKOnet* provides a closed-form optimal solution for linearly parametrized functionals, and, when applied to predict the evolution of cellular processes from real-world data, it achieves state-of-the-art accuracy at a fraction of the computational cost of all existing methods. Our methodology is based on the interpretation of diffusion processes as energy-minimizing trajectories in the probability space via the so-called JKO scheme, which we study via its first-order optimality conditions.

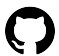 Source code: `https://github.com/antonioterpin/jkonet-star`

## 1 Introduction

Diffusion processes govern the homeostasis of biological systems [40], stem cells reprogramming [20, 36], and the learning dynamics of diffusion models [16, 22, 54] and transformers [19, 52]. The diffusion process of interest often originates from three quantities: a drift term due to a potential field, the interaction with other particles, and a stochastic term. If these three components are known, predictions follow from simple forward sampling [27] or the recent work in optimization in the probability space [1, 3, 11, 25, 34, 38, 41]. In this paper, we consider the case when the diffusion process is unknown, and we seek to learn its representation from observational data. The problem has been addressed when the trajectories of the individual particles are known [5, 32], but it is often the case that we only have "population data". For instance, single-cell RNA sequencing techniques enabled the collection of large quantities of data on biological systems [35], but the observer cannot access the trajectories of individual cells since measurements are destructive [20, 36]. The most promising avenue to circumvent the lack of particle trajectories is the Jordan-Kinderlehrer-Otto (JKO) scheme [24] which predicates that the particles as a whole move to decrease an aggregate energy, while not deviating too much from the current configuration. However, the JKO scheme entails an optimization problem in the probability space. Thus, the problem of finding the energy functional that minimizes a prediction error (w.r.t. observational data) takes the form of a computationally-challenging infinite-dimensional bilevel optimization problem, whereby the upper-level problem is the minimization of the prediction error and the lower-level problem is the JKO scheme. Recent work [1, 9] exploits the theory of optimal transport and in particular Brenier's theorem [7] to attack this bilevel optimization problem, a model henceforth referred to as JKOnet. Despite promising initial results in [9], this complexity undermines scalability, stability, and generality of the model.

38th Conference on Neural Information Processing Systems (NeurIPS 2024).

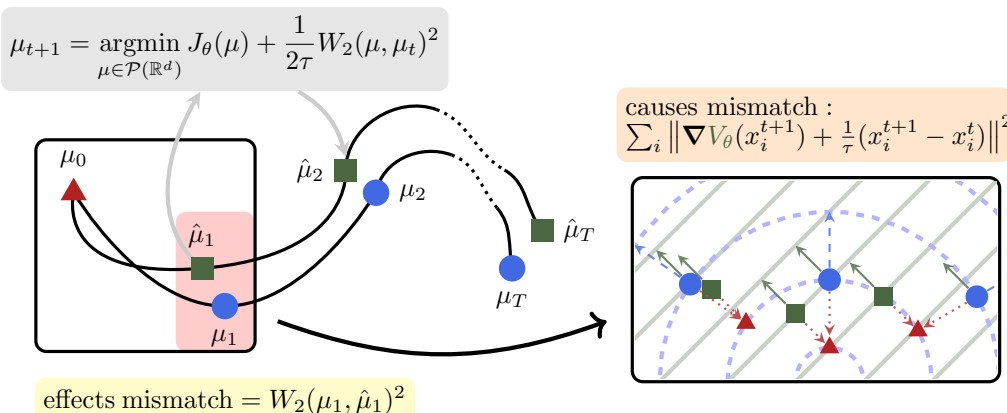

Figure 1: Given a sequence of snapshots $(\mu_0, \ldots, \mu_T)$ of a population of particles undergoing diffusion, we want to find the parameters $\theta$ of the parametrized energy function $J_\theta$ that best *explains* the particles evolution. Given $\theta$, the effects mismatch is the Wasserstein distance between the observed trajectory and the predicted trajectory obtained iteratively solving the JKO step with $J_\theta$. The first-order optimality condition in [30] applied to the JKO step suggests that the "gradient" of $J_\theta$ with respect to each $\hat{\mu}_t$ vanishes at optimality, i.e., for $\hat{\mu}_t = \mu_t$. For $J_\theta(\mu) = \int_{\mathbb{R}^d} V_\theta(x) \mathrm{d}\mu(x)$, this condition is depicted on the right. The gradient (dashed blue arrows) of the true $V$ (level curves in dashed blue) at each observed particle $x_i^{t+1}$ (blue circles) in the next snapshot $\mu_{t+1}$ opposes the displacement (dotted red arrows) from a particle $x_i^t$ (red triangles) in the previous snapshot $\mu_t$. Instead, the gradient (solid green arrows) of the estimated $V_\theta$ (level curves in solid green) at each observed particle $x_i^{t+1}$ (square) does not oppose the displacement from a particle $x_i^t$ in the previous snapshot $\mu_t$. This mismatch in the causes of the diffusion process is what JKOnet* minimizes.

Furthermore, to be practical, it is limited to learning only potential energies, modelling the underlying physics only partially. Alternatively, [10, 43] learn directly the transport map describing the evolution of the population (i.e., the effects), bypassing the representation of the underlying energy functional (i.e., the causes). Motivated by robustness, interpretability, and generalization, here we seek a method to learn the causes. In [23, 42], the authors try to learn a geometry that explains the observed transport maps. Unfortunately, the cost between two configurations along a cost-minimizing trajectory is often not a metric [47]. Other attempts include recurrent neural networks [21], neural ODEs [15], and Schrödinger bridges [12, 28].

**Contributions.** We study the first-order necessary optimality conditions for the JKO scheme, an optimization problem in the probability space, and show that these conditions can be exploited to learn the energy functional governing the underlying diffusion process from population data, effectively bypassing the complexity of the infinite-dimensional bilevel optimization problem. We provide a closed-form solution in the case of linearly parametrized energy functionals and a simple, interpretable, and efficient algorithm for non-linear parametrizations. Via exhaustive numerical experiments, we show that, in the case of potential energies only, JKOnet* outperforms the state-of-the-art in terms of solution quality, scalability, and computational efficiency and, in the until now unsolved case of general energy functionals, allows us to also learn interaction and internal energies that explain the observed population trajectories. When applied to predict the evolution of cellular processes, it achieves state-of-the-art accuracy at a fraction of the computational cost. Figure 1 shows an overview of our method, detailed in Section 3.

## 2 Diffusion processes via optimal transport

### 2.1 Preliminaries

The gradient of $\rho : \mathbb{R}^d \to \mathbb{R}$ is $\boldsymbol{\nabla}\rho \in \mathbb{R}^d$ and the Jacobian of $\phi : \mathbb{R}^d \to \mathbb{R}^n$ is $\boldsymbol{\nabla}\phi \in \mathbb{R}^{n \times d}$. We say that $f : \mathbb{R}^d \to \mathbb{R}$ has bounded Hessian if $\left\| \boldsymbol{\nabla}^2 f(x) \right\| \leq C$ for some $C > 0$ (and some matrix

norm $\|\cdot\|$). The divergence of $F : \mathbb{R}^d \to \mathbb{R}^n$ is $\boldsymbol{\nabla} \cdot F$ and its laplacian is $\nabla^2 F$. The identity function is $\mathrm{Id} : \mathbb{R}^d \to \mathbb{R}^d$, $\mathrm{Id}(x) = x$. We denote by $\mathcal{P}(\mathbb{R}^d)$ the space of (Borel) probability measures over $\mathbb{R}^d$ with finite second moment. For $\mu \in \mathcal{P}(\mathbb{R}^d)$, $\mathrm{supp}(\mu)$ is its support. The Dirac's delta measure at $x \in \mathbb{R}^d$, is $\delta_x$. All the functions are assumed to be Borel, and for $f : \mathbb{R}^d \to \mathbb{R}$, $\int_{\mathbb{R}^d} f(x)\mathrm{d}\mu(x)$ is the (Lebesgue) integral of $f$ w.r.t. $\mu$. If $\mu$ is absolutely continuous w.r.t. the Lebesgue measure, $\mu \ll \mathrm{d}x$, then it admits a density $\rho : \mathbb{R}^d \to \mathbb{R}_{\geq 0}$, and the integral becomes $\int_{\mathbb{R}^d} f(x)\rho(x)\mathrm{d}x$. The pushforward of $\mu$ via a (Borel) map $f : \mathbb{R}^d \to \mathbb{R}^d$ is the probability measure $f_\#\mu$ defined by $(f_\#\mu)(B) = \mu(f^{-1}(B))$; when $\mu$ is empirical with $N$, $\mu = \frac{1}{N}\sum_{i=1}^N \delta_{x_i}$, then $f_\#\mu = \frac{1}{N}\sum_{i=1}^N \delta_{f(x_i)}$. Given $\mu, \nu \in \mathcal{P}(\mathbb{R}^d)$, we say that a probability measure $\gamma \in \mathcal{P}(\mathbb{R}^d \times \mathbb{R}^d)$ is a transport plan (or coupling) between $\mu$ and $\nu$ if its marginals are $\mu$ and $\nu$. We denote the set of transport plans between $\mu$ and $\nu$ by $\Gamma(\mu, \nu)$. The Wasserstein distance between $\mu$ and $\nu$ is

$$W_2(\mu, \nu) := \left( \min_{\gamma \in \Gamma(\mu,\nu)} \int_{\mathbb{R}^d \times \mathbb{R}^d} \|x - y\|^2 \mathrm{d}\gamma(x, y) \right)^{\frac{1}{2}}. \tag{1}$$

When $\mu$ and $\nu$ are discrete, (1) is a linear program. If, additionally, they have the same number of particles, the optimal transport plan is $\gamma = (\mathrm{Id}, T)_\#\mu$ for some (transport) map $T : \mathbb{R}^d \to \mathbb{R}^d$ [39]. When $\mu$ is absolutely continuous, $\gamma = (\mathrm{Id}, \boldsymbol{\nabla}\psi)_\#\mu$ for some convex function $\psi$ [7].

## 2.2 The JKO scheme

Many continuous-time diffusion processes can be modeled by partial differential equations (PDEs) or stochastic differential equations (SDEs):

**Example 2.1** (Fokker-Planck). *The Fokker-Planck equation,*

$$\frac{\partial \rho(t, x)}{\partial t} = \boldsymbol{\nabla} \cdot (\boldsymbol{\nabla} V(x)\rho(t, x)) + \beta\nabla^2\rho(t, x), \tag{2}$$

*describes the time evolution of the distribution $\rho$ of a set of particles undergoing drift and diffusion,*

$$\mathrm{d}X(t) = -\boldsymbol{\nabla} V(X(t))\mathrm{d}t + \sqrt{2\beta}\mathrm{d}W(t),$$

*where $X(t)$ is the state of the particle, $V(x)$ the driving potential, and $W(t)$ the Wiener process.*

The pioneering work of Jordan, Kinderlehrer, and Otto [24], related diffusion processes to energy-minimizing trajectories in the Wasserstein space (i.e., probability space endowed with the Wasserstein distance), providing a discrete-time counterpart of the diffusion process, the JKO scheme,

$$\mu_{t+1} = \underset{\mu \in \mathcal{P}(\mathbb{R}^d)}{\mathrm{argmin}} \, J(\mu) + \frac{1}{2\tau}W_2(\mu, \mu_t)^2, \tag{3}$$

where $J : \mathcal{P}(\mathbb{R}^d) \to \mathbb{R} \cup \{+\infty\}$ is an energy functional and $\tau > 0$ is the time discretization.

**Example 2.2** (Fokker-Plank as a Wasserstein gradient flow). *The Fokker-Plank equation* (2) *results from the continuous-time limit (i.e., $\tau \to 0$) of the JKO scheme* (3) *for the energy functional*

$$J(\mu) = \int_{\mathbb{R}^d} V(x)\mathrm{d}\mu(x) + \beta\int_{\mathbb{R}^d} \rho(x)\log(\rho(x))\mathrm{d}x \quad \text{with} \quad \mathrm{d}\mu(x) = \rho(x)\mathrm{d}x.$$

## 2.3 Challenges

Section 2.2 suggests that we can interpret the problem of learning diffusion processes as the problem of learning the energy functional $J$ in (3). Specifically, the setting is as follows: We have access to sample populations $\mu_0, \mu_1, \ldots, \mu_T$, and we want to learn the energy functional governing their dynamics. A direct approach to tackle the inverse problem is a bilevel optimization, used, among others, for the model JKOnet in [9]. This approach bases on the following two facts. First, by Brenier's theorem, the solution of (3), $\mu_{t+1}$, can be expressed[1] as the pushforward of $\mu_t$ via the gradient of a convex function $\psi_t : \mathbb{R}^d \to \mathbb{R}$ and, thus,

$$W_2(\mu_t, \mu_{t+1})^2 = \int_{\mathbb{R}} \|x - \boldsymbol{\nabla}\psi_t(x)\|^2\mathrm{d}\mu_t(x).$$

---

[1]Under an absolute continuity assumption.

Second, the optimization problem (3) is equivalently written as

$$\underset{\psi_t \in C}{\operatorname{argmin}} \, J(\boldsymbol{\nabla}\psi_{t\#}\mu_t) + \frac{1}{2\tau} \int_{\mathbb{R}^d} \|x - \boldsymbol{\nabla}\psi_t(x)\|^2 \mathrm{d}\mu_t(x),$$

where $C$ is the class of continuously differentiable convex functions from $\mathbb{R}^d$ to $\mathbb{R}$. Therefore, the learning task can be cast into the following bilevel optimization problem, which minimizes the discrepancy between the observations ($\mu_t$) and the predictions of the model ($\hat{\mu}_t$):

$$
\begin{aligned}
&\min_J \sum_{t=1}^{T} W_2(\hat{\mu}_t, \mu_t)^2 \\
&\text{s.t. } \hat{\mu}_0 = \mu_0, \quad \hat{\mu}_{t+1} = \boldsymbol{\nabla}\psi_t^*\hat{\mu}_t, \\
&\quad \psi_t^* := \underset{\psi \in C}{\operatorname{argmin}} \, J(\boldsymbol{\nabla}\psi_{t\#}\hat{\mu}_t) + \frac{1}{2\tau} \int_{\mathbb{R}^d} \|x - \boldsymbol{\nabla}\psi_t(x)\|^2 \mathrm{d}\hat{\mu}_t(x).
\end{aligned}
\tag{4}
$$

A practical implementation of the above requires a parametrization of both the transport map and the energy functional. The former problem has been tackled via input convex neural network (ICNN) parametrizing $\psi_t$ [2, 8] or via the "Monge gap" [51]. The second problem is only addressed for energy functional of the form $J(\mu) = \int_{\mathbb{R}} V_\theta(x) \mathrm{d}\mu(x)$, without interaction and internal energies, where $V_\theta$ is a non-linear function approximator [9].

> **Challenges.** This approach suffers from two major limitations. First, bilevel optimization problems are notoriously hard and we should therefore expect (4) to be computationally challenging. Second, most energy functionals are not potential energies but include interactions and internal energy terms as well. Although it is tempting to include other terms in the energy functional $J$ (e.g., parametrizing interaction and internal energies), the complexity of the bilevel optimization problem renders such an avenue viable only in principle.

## 3 Learning diffusion at lightspeed

Our methodology consists of replacing the optimization problem (3) with its first-order necessary conditions for optimality. This way, we bypass its computational complexity which, ultimately, leads to the bilevel optimization problem (4). Perhaps interestingly, our methodology for learning diffusion is based on first principles: whereas e.g. [9] minimizes an error on the *effects* (the predictions), we minimize an error on the *causes* (the energy functionals driving the diffusion process); see Figure 1. As we detail in Section 4, the resulting learning algorithms are significantly faster and more effective.

### 3.1 Intuition

To start, we illustrate our idea in the Euclidean case (i.e., $\mathbb{R}^d$) and later generalize it to the probability space (i.e., $\mathcal{P}(\mathbb{R}^d)$). Consider the problem of learning the energy functional $J : \mathbb{R}^d \to \mathbb{R} \cup \{+\infty\}$ of the analog of the JKO scheme in the Euclidean space, the proximal operator

$$x_{t+1} = \underset{x \in \mathbb{R}^d}{\operatorname{argmin}} \, J(x) + \frac{1}{2\tau}\|x - x_t\|^2. \tag{5}$$

Under sufficient regularity, we can replace (5) by its first-order optimality condition

$$\boldsymbol{\nabla}J(x_{t+1}) + \frac{1}{\tau}(x_{t+1} - x_t) = 0. \tag{6}$$

Given a dataset $(x_0, x_1, \ldots, x_T)$, we seek the energy functional that best fits the optimality condition:

$$\min_J \sum_{t=0}^{T-1} \left\| \boldsymbol{\nabla}J(x_{t+1}) + \frac{1}{\tau}(x_{t+1} - x_t) \right\|^2. \tag{7}$$

In the probability space, we can proceed analogously and replace (3) with its first-order optimality conditions. This analysis, which is based on recent advancements in optimization in the probability space [30], allows us to formulate the learning task as a single-level optimization problem.

## 3.2 Potential energy

Consider initially the case where the energy functional is a potential energy, for $V : \mathbb{R}^d \to \mathbb{R}$,

$$J(\mu) = \int_{\mathbb{R}^d} V(x)\mathrm{d}\mu(x). \tag{8}$$

The following proposition is the counterpart of (6) in $\mathcal{P}(\mathbb{R}^d)$:

**Proposition 3.1** (Potential energy). *Assume $V$ is continuously differentiable, lower bounded, and has a bounded Hessian. Then, the JKO scheme* (3) *has an optimal solution $\mu_{t+1}$ and, if $\mu_{t+1}$ is optimal for* (3)*, then there is an optimal transport plan $\gamma_t$ between $\mu_t$ and $\mu_{t+1}$ such that*

$$\int_{\mathbb{R}^d \times \mathbb{R}^d} \left\| \boldsymbol{\nabla} V(x_{t+1}) + \frac{1}{\tau}(x_{t+1} - x_t) \right\|^2 \mathrm{d}\gamma_t(x_t, x_{t+1}) = 0.$$

Proposition 3.1 is by all means the analog of (6), since for the integral to be zero, $\boldsymbol{\nabla} V(x_{t+1}) + \frac{1}{\tau}(x_{t+1} - x_t) = 0$ must hold for all $(x_t, x_{t+1}) \in \mathrm{supp}(\gamma_t)$. Since the collected population data $\mu_0, \mu_1, \ldots, \mu_T$ are not optimization variables in the learning task, the optimal transport plan $\gamma_t$ can be computed beforehand. Thus, we can learn the energy functional representation by minimizing over a class of continuously differentiable potential energy functions the loss function

$$\sum_{t=0}^{T-1} \int_{\mathbb{R}^d \times \mathbb{R}^d} \left\| \boldsymbol{\nabla} V(x_{t+1}) + \frac{1}{\tau}(x_{t+1} - x_t) \right\|^2 \mathrm{d}\gamma_t(x_t, x_{t+1}). \tag{9}$$

## 3.3 Arbitrary energy functionals

Consider now the general case where the energy functional consists of a potential energy (with the potential function $V : \mathbb{R}^d \to \mathbb{R}$), interaction energy (with interaction kernel $U : \mathbb{R}^d \to \mathbb{R}$), and internal energy (expressed as the entropy weighted by $\beta \in \mathbb{R}_{\geq 0}$):

$$J(\mu) = \int_{\mathbb{R}^d} V(x)\mathrm{d}\mu(x) + \int_{\mathbb{R}^d \times \mathbb{R}^d} U(x-y)\mathrm{d}(\mu \times \mu)(x, y) + \beta \int_{\mathbb{R}^d} \rho(x) \log(\rho(x))\mathrm{d}x. \tag{10}$$

The first-order necessary optimality condition for the JKO scheme then reads as follows.

**Proposition 3.2** (General case). *Assume $V$ and $U$ are continuously differentiable, lower bounded, and have a bounded Hessian. Then, the JKO scheme* (3) *has an optimal solution $\mu_{t+1}$ which is absolutely continuous with density $\rho_{t+1}$ and, if $\mathrm{d}\mu_{t+1}(x) = \rho_{t+1}(x)\mathrm{d}x$ is optimal for* (3)*, then there is an optimal transport plan $\gamma_t$ between $\mu_t$ and $\mu_{t+1}$ such that*

$$0 = \int_{\mathbb{R}^d \times \mathbb{R}^d} \left\| \boldsymbol{\nabla} V(x_{t+1}) + \int_{\mathbb{R}^d} \boldsymbol{\nabla} U(x_{t+1} - x'_{t+1})\mathrm{d}\mu_{t+1}(x'_{t+1}) \right.$$

$$\left. + \beta \frac{\boldsymbol{\nabla} \rho_{t+1}(x_{t+1})}{\rho_{t+1}(x_{t+1})} + \frac{1}{\tau}(x_{t+1} - x_t) \right\|^2 \mathrm{d}\gamma_t(x_t, x_{t+1}).$$

Thus, Proposition 3.2 suggests that the energy functional can be learned by minimizing over a class of continuously differentiable potential and internal energy functions and $\beta \in \mathbb{R}_{\geq 0}$ the loss function

$$\sum_{t=0}^{T-1} \int_{\mathbb{R}^d \times \mathbb{R}^d} \left\| \boldsymbol{\nabla} V(x_{t+1}) + \int_{\mathbb{R}^d} \boldsymbol{\nabla} U(x_{t+1} - x'_{t+1})\mathrm{d}\mu_{t+1}(x'_{t+1}) \right.$$

$$\left. + \beta \frac{\boldsymbol{\nabla} \rho_{t+1}(x_{t+1})}{\rho_{t+1}(x_{t+1})} + \frac{1}{\tau}(x_{t+1} - x_t) \right\|^2 \mathrm{d}\gamma_t(x_t, x_{t+1}). \tag{11}$$

*Remark* 3.3. We generalize the setting to time-varying energies in Section 4.4 and in Appendix B.

## 3.4 Parametrizations

For our model JKOnet*, we parametrize the energy functional at a measure $\mathrm{d}\mu = \rho(x)\mathrm{d}x$ as follows:

$$J_\theta(\mu) = \int_{\mathbb{R}^d} V_{\theta_1}(x)\mathrm{d}\mu(x) + \int_{\mathbb{R}^d \times \mathbb{R}^d} U_{\theta_2}(x-y)\mathrm{d}(\mu \times \mu)(x, y) + \theta_3 \int_{\mathbb{R}^d} \rho(x) \log(\rho(x))\mathrm{d}x,$$

where $\theta_1, \theta_2 \in \mathbb{R}^n, \theta_3 \in \mathbb{R}$, and we set $\theta = \left[\theta_1^\top, \theta_2^\top, \theta_3^\top\right]^\top \in \mathbb{R}^{2n+1}$.

| Model | FLOPS per epoch | Seq. op. per particle | Generality $V(x)$ | $\beta$ | $U(x)$ |
|---|---|---|---|---|---|
| JKOnet w/o TF | $\mathcal{O}\left(T\left(DNd + \frac{N^2 \log(N)}{\varepsilon^2}\right)\right)$ | $\mathcal{O}\left(T\left(DNd + \frac{N^2 \log(N)}{\varepsilon^2}\right)\right)$ | ✓ | ✗ | ✗ |
| JKOnet w/ TF | $\mathcal{O}\left(T\left(DNd + \frac{N^2 \log(N)}{\varepsilon^2}\right)\right)$ | $\mathcal{O}\left(DNd + \frac{N^2 \log(N)}{\varepsilon^2}\right)$ | ✓ | ✗ | ✗ |
| JKOnet w/ MG w/o TF | $\mathcal{O}\left(TD\left(Nd + \frac{N^2 \log(N)}{\varepsilon^2}\right)\right)$ | $\mathcal{O}\left(TD\left(Nd + \frac{N^2 \log(N)}{\varepsilon^2}\right)\right)$ | ✓ | ✗ | ✗ |
| JKOnet w/ MG, TF | $\mathcal{O}\left(TD\left(Nd + \frac{N^2 \log(N)}{\varepsilon^2}\right)\right)$ | $\mathcal{O}\left(D\left(Nd + \frac{N^2 \log(N)}{\varepsilon^2}\right)\right)$ | ✓ | ✗ | ✗ |
| JKOnet* w/o $U(x)$ | $\mathcal{O}\left(TNd\right)$ | $\mathcal{O}\left(d\right)$ | ✓ | ✓ | ✗ |
| JKOnet* w/ $U(x)$ | $\mathcal{O}\left(TN^2d\right)$ | $\mathcal{O}\left(Nd\right)$ | ✓ | ✓ | ✓ |
| JKOnet$_l^*$ w/o $U(x)$ | $\mathcal{O}\left(TNdn + n^3\right)$ | $\mathcal{O}\left(TNdn + n^3\right)$ | ✓ | ✓ | ✗ |
| JKOnet$_l^*$ w/ $U(x)$ | $\mathcal{O}\left(TN^2dn + n^3\right)$ | $\mathcal{O}\left(TN^2dn + n^3\right)$ | ✓ | ✓ | ✓ |

Table 1: Per-epoch complexity (in FLOPs) and per-particle minimum number of sequential operations (maximum parallelization) for the `JKOnet` and `JKOnet`$^*$ model families (we refer to the linear parametrization of our model with `JKOnet`$_l^*$). Here, $T$ is the length of the population trajectory, $N$ the number of particles in the snapshots of the population (assumed constant), $d$ is the dimensionality of the system, $n$ is the number of features for the linear parametrization, $D, \varepsilon,$ `TeacherForcing` (TF) are `JKOnet` parameters: the number of inner operations (which may or not be constant), the accuracy required for the Sinkhorn algorithm, and a training modality (see [9] for details), respectively. MG stands for the Monge gap regularization [51].

**Linear parametrizations.** When the parametrizations are linear, i.e. $V_{\theta_1}(x) = \theta_1^\top \phi(x)$, $U_{\theta_2}(x - y) = \theta_2^\top \phi(x-y)$ for feature maps $\phi_1, \phi_2 : \mathbb{R}^d \to \mathbb{R}^n$, the optimal $\theta^*$ can be computed in closed-form:

**Proposition 3.4.** *Assume that the features $\phi_{1,i}$ and $\phi_{2,i}$ are continuously differentiable, bounded, and have bounded Hessian. Define the matrix $y_t : \mathbb{R}^d \to \mathbb{R}^{(2n+1) \times d}$ by*

$$y_t(x_t) \coloneqq \left[\boldsymbol{\nabla}\phi_1(x_t)^\top, \int_{\mathbb{R}^d} \boldsymbol{\nabla}\phi_2(x_t - x_t')^\top \mathrm{d}\mu_t(x_t'), \frac{\boldsymbol{\nabla}\rho_t(x_t)}{\rho_t(x_t)}\right]^\top$$

*and suppose that the data is sufficiently exciting so that $\sum_{t=1}^T \int_{\mathbb{R}^d} y_t(x_t) y_t(x_t)^\top \mathrm{d}\mu_t(x_t)$ is invertible. Then, the optimal solution of (11) is*

$$\theta^* = \frac{1}{\tau}\left(\sum_{t=1}^T \int_{\mathbb{R}^d} y_t(x_t)y_t(x_t)^\top \mathrm{d}\mu_t(x_t)\right)^{-1}\left(\sum_{t=0}^{T-1} \int_{\mathbb{R}^d \times \mathbb{R}^d} y_t(x_{t+1})(x_{t+1} - x_t)\mathrm{d}\gamma_t(x_t, x_{t+1})\right).$$
(12)

*Remark* 3.5. The excitation assumption can be enforced via regularization terms $\lambda_i \|\theta_i\|^2$, with $\lambda_i > 0$, in the loss (11). Another practical alternative is to use pseudoinverse or solve the least-squares problem corresponding to (12) by gradient descent.

**Non-linear parametrizations.** When the parametrizations are non-linear, we minimize (11) by gradient descent.

**Inductive biases.** By fixing any of the parameters $\theta_1, \theta_2, \theta_3$ to zero, the corresponding energy term is dropped from the model. It is thus possible to inject into `JKOnet`$^*$ the proper inductive bias when additional information on the underlying diffusion process are known. For instance, if the process is deterministic and driven by an external potential, one can set $\theta_2 = \theta_3 = 0$. Similarly, if it can be assumed that the interaction between the particles is negligible, we can set $\theta_2 = 0$.

### 3.5 Why first-order conditions

Here, we motivate the theoretical benefits of `JKOnet`$^*$ over `JKOnet` using the desiderata:

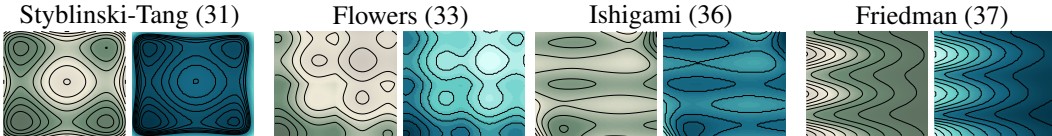

Figure 2: Level curves of the true (green-colored) and estimated (blue-colored) potentials (31), (33), (36) and (37), see Appendix F. See also Figure 6 in Appendix A.

1. Total computational complexity per epoch (i.e., the cost to process the observed populations).
2. Per-particle computational complexity (i.e., the cost to process a single particle when maximally parallelizing the algorithm, prior to merging the results).
3. Representational capacity of the method (i.e., which energy terms the model can learn).

We collect this analysis in Table 1. Fundamentally, the first-order optimality conditions allow a reformulation of the learning problem that decouples prediction of the population evolution and learning of the dynamics (such coupling is the crux of (4)). As a result, JKOnet* enjoys higher parallelizability. We also observe that the interaction energy comes with an increase in complexity, and in a way resembles the attention mechanisms in transformers [19, 52]. The linear dependence of JKOnet* on the size of the batch implies that our method can process larger batch sizes for free (to process the entire dataset we need fewer steps in an inverse relationship with the batch size). In practice, this actually increases the speed (less memory swaps). These considerations do not hold for the JKOnet family. Finally, JKOnet* enjoys enhanced representational power and interpretability. JKOnet$_l^*$ generally needs more computation per epoch (primarily related to the number of features) but requires a single epoch. In Table 1 we also report the computational complexity of the variants of JKOnet using a vanilla multi-layer perceptron (MLP) with Monge gap regularization [51] instead of a ICNN as a parametrization of the transport map. Despite the success in simplifying the training of transport maps over the use of ICNN [51], the Monge gap requires the solution of an optimal transport problem at every inner iteration, an unbearable slowdown [33].

*Remark* 3.6. Unlike JKOnet, JKOnet* requires the construction of the optimal transport couplings beforehand. However, JKOnet constructs a new optimal transport plan at each iteration depending on the current estimate of the potential, whereas JKOnet* needs to do so only once, at the beginning. Moreover, as discussed in Section 4.1, in Section 4.2, in the application to single-cell diffusion dynamics in Section 4.4, and in the ablations in Appendix C.2, this additional cost is minimal.

## 4 Experiments

The code for the experiments is available at https://github.com/antonioterpin/jkonet-star. We include the training and architectural details for the JKOnet* models family in Appendix C. The settings of the baselines considered are the one provided by the corresponding papers, reported for completeness in Appendix E, and the hardware setup is described in Appendix C.7. In all the experiments, we allow the models a budget of 1000 epochs.

**Our models.** We use the following terminology for our method. JKOnet* is the most general non-linear parametrization in Section 3.4 and JKOnet$_V^*$ introduces the inductive bias $\theta_2 = \theta_3 = 0$. Similarly, we refer to the linear parametrizations by JKOnet$_{l,V}^*$ and JKOnet$_l^*$.

**Metrics.** To evaluate the prediction capabilities we use the one-step-ahead earth-mover distance (EMD), $\min_{\gamma \in \Gamma(\mu_t, \hat{\mu}_t)} \int_{\mathbb{R}^d \times \mathbb{R}^d} \|x - y\| \mathrm{d}\gamma(x, y)$, where $\mu_t$ and $\hat{\mu}_t$ are the observed and predicted populations. In particular, we consider the average and standard deviation over a trajectory.

### 4.1 Training at lightspeed

**Experimental setting.** We validate the observations in Section 3.5 comparing (i) the EMD error, (ii) the convergence ratio, and (iii) the time per epoch required by the different methods on a synthetic dataset (see Appendix B) consisting of particles subject to a non-linear drift, $x_{t+1} = x_t - \tau \boldsymbol{\nabla} V(x_t)$, with $\tau = 0.01$, $T = 5$, and the potential functions $V(x)$ (31)-(45) in Appendix F, shown in Figure 2 and in Figure 6 in Appendix A.

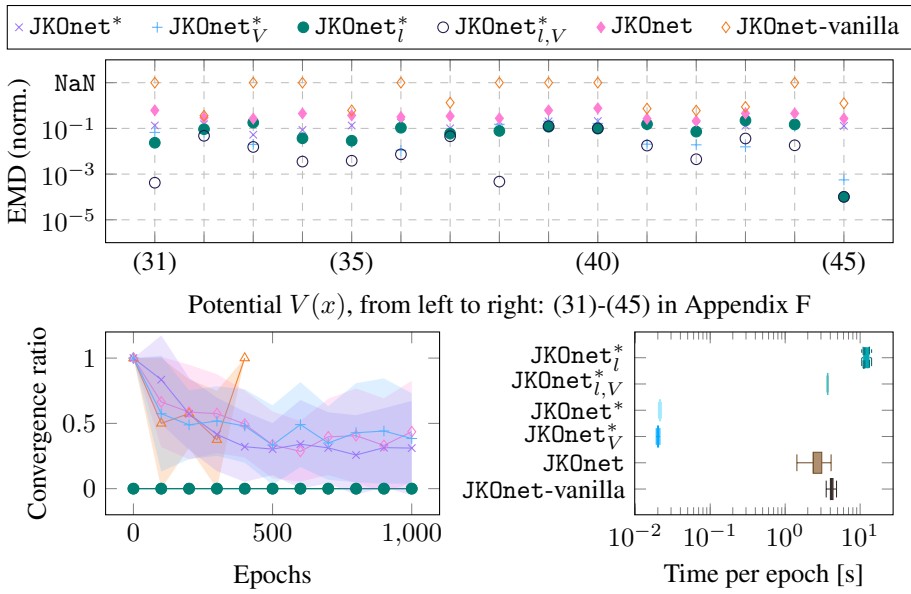

Figure 3: Numerical results of Section 4.1. The scatter plot displays points $(x_i, y_i)$ where $x_i$ indexes the potentials in Appendix F and $y_i$ are the errors (EMD, normalized so that the maximum error among all models and all potentials is 1) obtained with the different models. We mark with `NaN` each method that has diverged during training. The plot on the bottom-left shows the EMD error trajectory during training (normalized such that 0 and 1 are the minimum and maximum EMD), averaged over all the experiments. The shaded area represents the standard deviation. The box plot analyses the time per epoch required by each method. The statistics are across all epochs and all potential energies.

**Results.** Figure 3 summarizes our results. All our methods perform uniformly better than the baselines, regardless of the generality. The speed improvement of the `JKOnet*` models family suggests that a theoretically guided loss may provide strong computational benefits on par with sophisticated model architectures. Our linearly parametrized models, `JKOnet`$_l^*$ and `JKOnet`$_{l,V}^*$, require a computational time per epoch comparable to the `JKOnet` family, but they only need one epoch to solve the problem optimally. Our non-linear models, `JKOnet`$^*$ and `JKOnet`$_V^*$, instead both require significantly lower time per epoch and converge faster than the `JKOnet` family. In these experiments, the computational cost associated with the optimal transport plans beforehand amounts to as little as $0.03 \pm 0.01$s, and thus has negligible impact on training time. The true and estimated level curves of the potentials are depicted in Figure 2 and Figure 6 in Appendix A. Compared to `JKOnet`, our model also requires a simpler architecture: we drop the additional ICNN used in the inner iteration and the related training details (e.g., the strong convexity regularizer and the teacher forcing). Notice that simply replacing the ICNN in `JKOnet` with a vanilla MLP deprives the method of the theoretical connections with optimal transport, which, in our experiments, appears to be associated with stability (`NaN` in Figure 3).

The results suggest orders of magnitude of improvement also in terms of accuracy of the predictions. These performance gains can be observed also between the linear and non-linear parametrization of `JKOnet`$^*$. In view of Proposition 3.4, this is not unexpected: the linear parametrization solves the problem optimally, when the features are representative enough. However, the feature selection presents a problem in itself; see e.g. [4, §3 and §4]. Thus, whenever applicable, we invite researchers and practitioners to adopt the linear parametrization, and the non-linear parametrization as demanded by the dimensionality of the problem. We further discuss the known failure modes in Appendix G.

## 4.2 Scaling laws

**Experimental setting.** We assess the performance of `JKOnet`$_V^*$ to recover the correct potential energy given $N \in \{1000, 2500, 5000, 7500, 10000\}$ particles in dimension $d \in \{10, 20, 30, 40, 50\}$, generated as in Section 4.1.

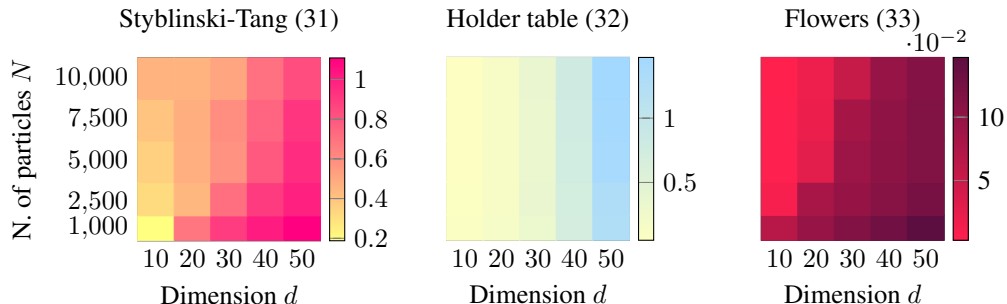

Figure 4: Numerical results of Section 4.2, reported in full in Figure 7 in Appendix A. The colors represent the EMD error, which appears to scale sublinearly with the dimension $d$.

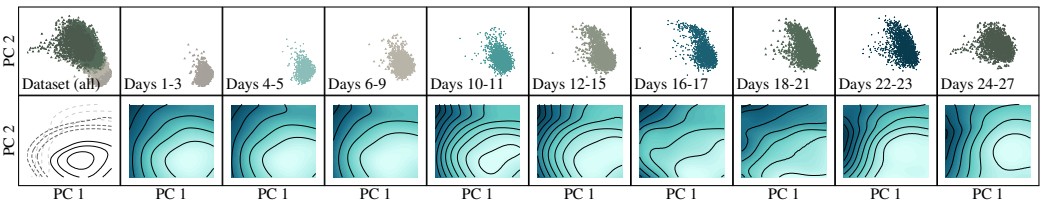

Figure 5: Visualizations of Section 4.4. The top row shows the two principal components of the scRNA-seq data, ground truth (green, days 1-3, 6-9, 12-15, 18-21, 24-27) and interpolated (blue, days 4-5, 10-11, 16-17, 22-23). The bottom row displays the estimated potential level curves over time. The bottom left plot superimposes the same three level curves for days 1-3 (solid), 12-15 (dashed), and 24-27 (dashed with larger spaces) to highlight the time-dependency.

**Results.** We summarize our findings in Figure 4 for the potentials (31)-(33) and in Figure 7 in Appendix A for all other the potentials. Since the EMD error is related to the Euclidean norm, it is expected to grow linearly with the dimension $d$ (i.e., along the rows); here, the growth is sublinear up to the point where the number of particles is not informative enough: along the columns, the error decreases again. The time complexity of the computation of the optimal transport plans is influenced linearly by the dimensionality $d$, and is negligible compared to the solution of the linear program, which depends only on the number of particles; we further discuss these effects in Appendix C.2. We thus conclude that JKOnet$^*$ is well suited for high-dimensional tasks.

### 4.3 General energy functionals

**Experimental setting.** We showcase the capabilities of the JKOnet$^*$ models to recover the potential, interaction, and internal energies selected as combinations of the functions in Appendix F[2] and noise levels $\beta \in \{0.0, 0.1, 0.2\}$. To our knowledge, this is the first model to recover all three energy terms.

**Results.** We summarize our findings on the right. Compared to the setting in Section 4.1, there are two additional sources of inaccuracies: (i) the noise, which introduces an inevitable sampling error, and the (ii) the estimation of the densities (see Appendix C for training details). Nonetheless, the low EMD errors demonstrate the capability of JKOnet$^*$ to recover the energy components that best explain the observed populations.

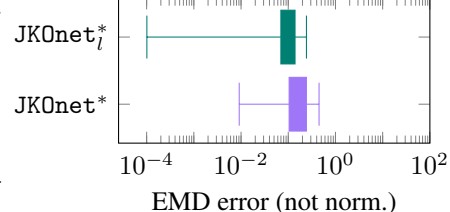

### 4.4 Learning single-cell diffusion dynamics

**Experimental setting.** Understanding the time evolution of cellular processes subject to external stimuli is a fundamental open question in biology. Motivated by the intuition that cells differentiate minimizing some energy functional, we deploy JKOnet$^*$ to analyze the embryoid body single-cell

---

[2]We exclude the functional (32) from the interaction energies due to numerical issues in the data generation.

RNA sequencing (scRNA-seq) data [35] describing the differentiation of human embryonic stem cells over a period of 27 days. We follow the data pre-processing in [50, 49]; in particular, we use the same processed artifacts of the embryoid data, which contains the first 100 components of the principal components analysis (PCA) of the data and, following [49], we focus on the first five. The cells are sequenced in five snapshots (days 1-3, 6-9, 12-15, 18-21, 24-27); we visualize the first two principal components in Figure 5. The visualization suggests that the energy governing the evolution is time-varying, possibly due to unobserved factors. For this, we condition the non-linear parametrization in Section 3 on time $t \in \mathbb{R}$ and minimize the loss

$$\sum_{t=0}^{T-1} \int_{\mathbb{R}^d \times \mathbb{R}^d} \left\| \boldsymbol{\nabla} V(x_{t+1}, t+1) + \frac{1}{\tau}(x_{t+1} - x_t) \right\|^2 \mathrm{d}\gamma_t(x_t, x_{t+1}).$$

To predict the evolution of the particles, then, we use the implicit scheme (see Appendix B)

$$x_{t+1} = x_t - \tau \boldsymbol{\nabla} V(x_{t+1}, t+1).$$

We train the time-varying extension of JKOnet$_V^*$, JKOnet and JKOnet-vanilla for 100 epochs on 60% of the data at each time and we compute the EMD between the observed $\mu_t$ (40% remaining data) and one-step ahead prediction $\hat{\mu}_t$ at each timestep. We then average over the trajectory and report the statistics for 5 seeds.

**Results.** We display the time evolution of the first two principal components of the level curves of the inferred potential energy in Figure 5, along with the cells trajectory (in green the data, in blue the interpolated predictions). As indicated by the table on the right, JKOnet* outperforms JKOnet. We also compare JKOnet* with recent work in the literature which focuses on the slightly different setting, namely the inference of $\mu_t$ from the evolution at all other time steps $\mu_k$, $k \neq t$, without train/test split of the data (the numerical values are taken directly from [12, 49] and our statistics are computed over the timesteps). Since the experimental setting slightly differs, we limit ourselves to observe that JKOnet* achieves state-of-the-art performance, but with significantly lower training time: JKOnet* trains in a few minutes, while the methods listed take hours to run. We further discuss these results in Appendix E.

| Algorithm | EMD |
|---|---|
| JKOnet [9] | $1.363 \pm 0.214$ |
| JKOnet-vanilla [9] | $3.237 \pm 1.135$ |
| TrajectoryNet [50] | $0.848 \pm --$ |
| Reg. CNF [17] | $0.825 \pm --$ |
| DSB [14] | $0.862 \pm 0.023$ |
| I-CFM [49] | $0.872 \pm 0.087$ |
| SB-CFM [49] | $1.221 \pm 0.380$ |
| OT-CFM [49] | $0.790 \pm 0.068$ |
| NLSB [28] | $0.74 \pm --$ |
| MIOFLOW [23] | $0.79 \pm --$ |
| DMSB [12] | $0.67 \pm --$ |
| JKOnet$_V^*$ (time-varying) | $0.624 \pm 0.007$ |

## 5  Conclusion and limitations

**Contributions.** We introduced JKOnet*, a model which recovers the energy functionals governing various classes of diffusion processes. The model is based on the novel study of the first-order optimality conditions of the JKO scheme, which drastically simplifies the learning task. In particular, we replace the complex bilevel optimization problem with a simple mean-square error, outperforming existing methods in terms of computational cost, solution accuracy, and expressiveness. In the prediction of cellular processes, JKOnet* achieves state-of-the-art performance and trains in less than a minute, compared to the hours of all existing methods.

**Limitations.** Our work did not address a few important challenges, which we believe to be exciting open questions. On the practical side, JKOnet* owns its performances to a loss function motivated by deep theoretical results. However, its architecture is still "vanilla" and we did not investigate data domains like images. Moreover, this work does not investigate in detail the choice of features for the linear parametrization, which in our experiments displays extremely promising results nonetheless. We further discuss the known failure modes in Appendix G.

**Outlook.** We expect the approach followed in this work to apply to other exciting avenues of applied machine learning research, such as population steering [47], reinforcement learning [37, 44, 48], diffusion models [16, 22, 54] and transformers [19, 52].

## Acknowledgments and Disclosure of Funding

We thank Elisabetta Fedele for the helpful and detailed comments. We also thank the reviewers for their constructive suggestions. This project has received funding from the Swiss National Science Foundation under the NCCR Automation (grant agreement 51NF40_180545).

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

## A    Eye candies

We collect the level curves of the ground-truth potentials and the ones recovered with each of our methods, together with the predictions of the particles evolution super-imposed to the ground-truth population data, in Figure 6. The plots discussed in Section 4.2 are in Figure 7.

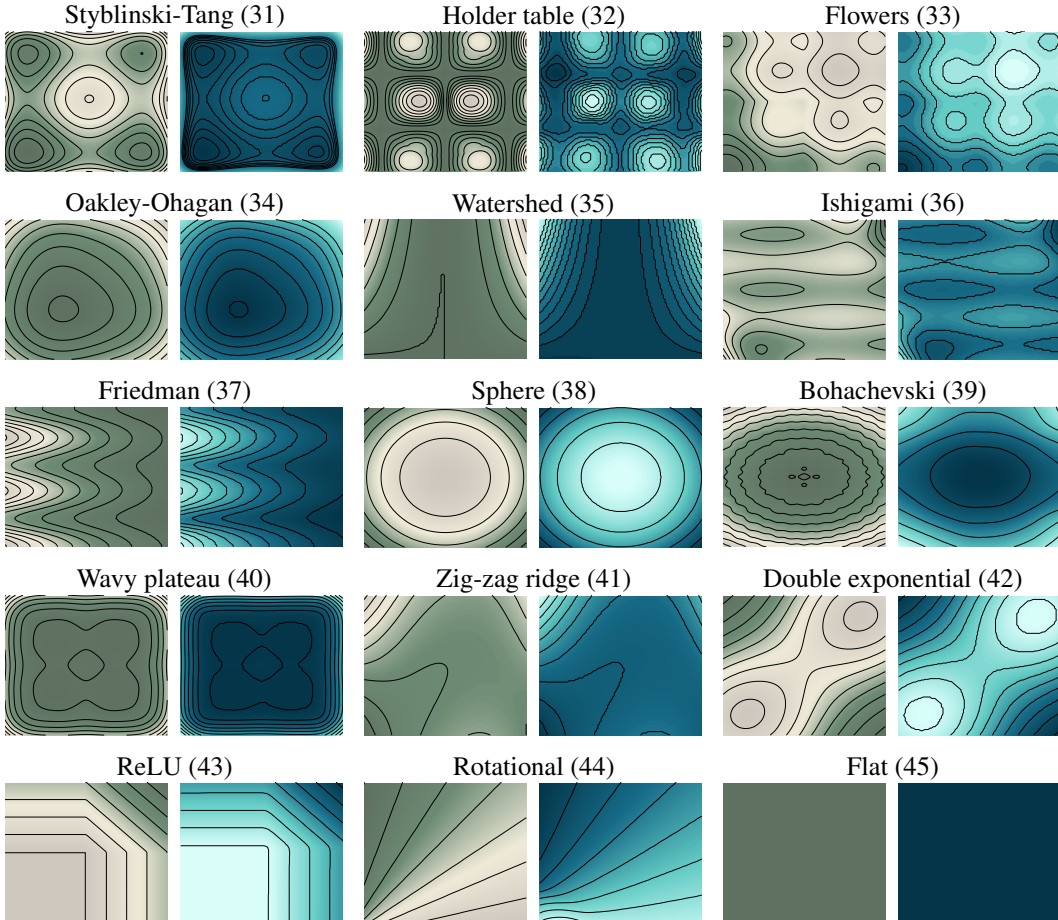

Figure 6: Level curves of the true (green-colored) and estimated (blue-colored) potentials in Appendix F, from top-left to bottom-right, row-by-row.

## B    Details on the data generation and prediction of the particles evolution

The prediction scheme for each particle,

$$x_{t+1} = x_t - \tau \boldsymbol{\nabla} V(x_t) - \tau \int_{\mathbb{R}^d} \boldsymbol{\nabla} U(x_t - y) \mathrm{d}\mu_t(y) + \sqrt{2\tau\beta} n_t, \tag{13}$$

where $n_t$ is sampled from the $d$-dimensional Gaussian distribution at each $t$, follows from the Euler-Maruyama discretization (see [27]) of the diffusion process

$$\mathrm{d}X(t) = -\boldsymbol{\nabla} V(X(t))\mathrm{d}t - \int_{\mathbb{R}^d} \boldsymbol{\nabla} U(X(t) - y)\mathrm{d}\mu_t(y)\mathrm{d}t + \sqrt{2\beta}\mathrm{d}W(t). \tag{14}$$

In particular, we sample $2N$ particles (in Section 4.1, $N = 1000$; in Section 4.2, $N$ is indicated in the experiment) uniformly in $[-4, 4]^d$ and update their state according to (13) for 5 timesteps, for a total of 6 snapshots including the initialization. Then, the data of the first $N$ particles is used for training, and the data of the remaining $N$ particles is left out for testing.

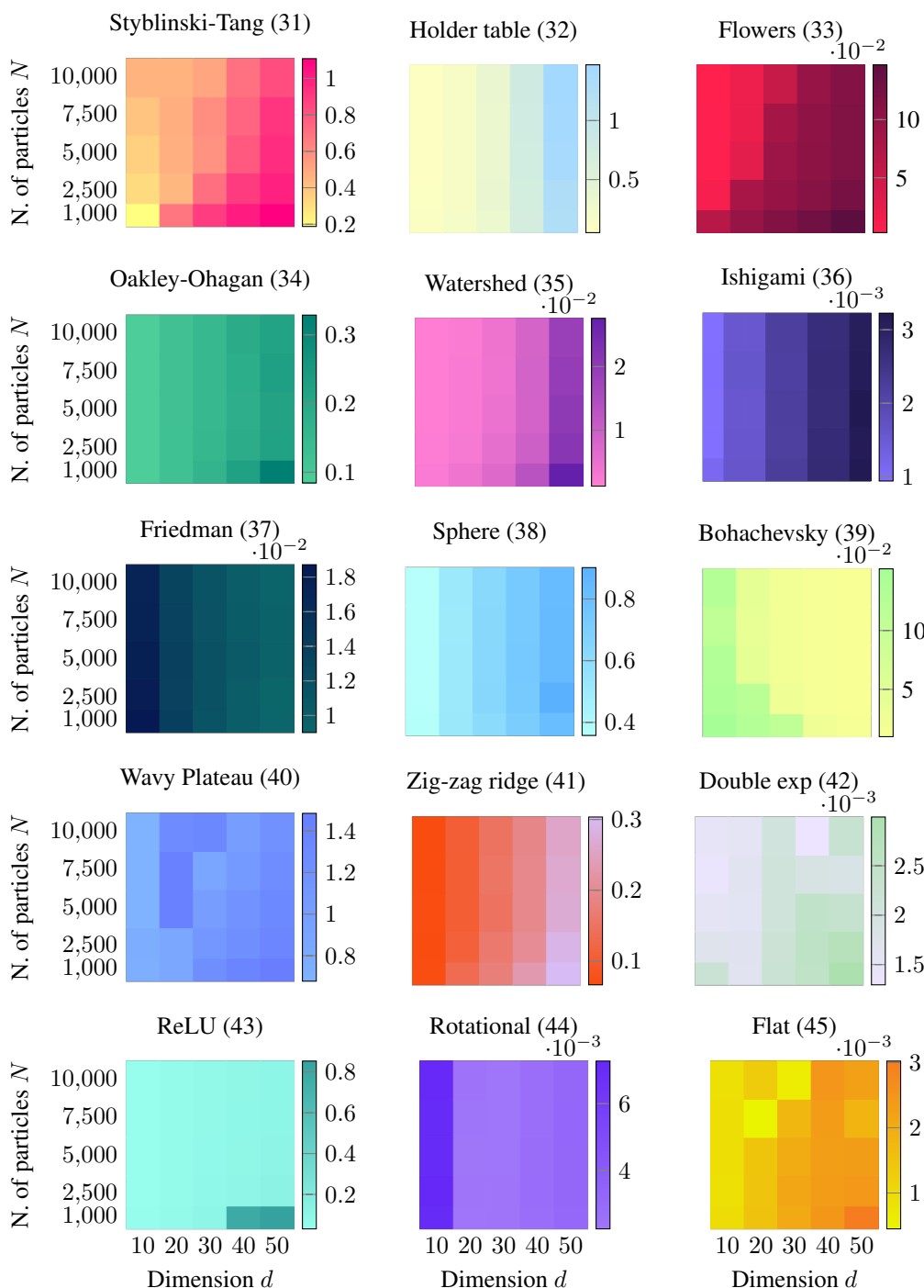

Figure 7: Additional numerical results for Section 4.2. Each heat-map corresponds to a functional in Appendix F, from top-left to bottom-right, row-by-row. The $x$-axis corresponds to the dimension and the $y$-axis corresponds to the number of particles. The colors represent the EMD error. Thus, a method that scales well to high-dimensional settings should display a relatively stable color along the rows: the error is related to the norm and, thus, is linear in the dimension $d$; here, the growth is sublinear.

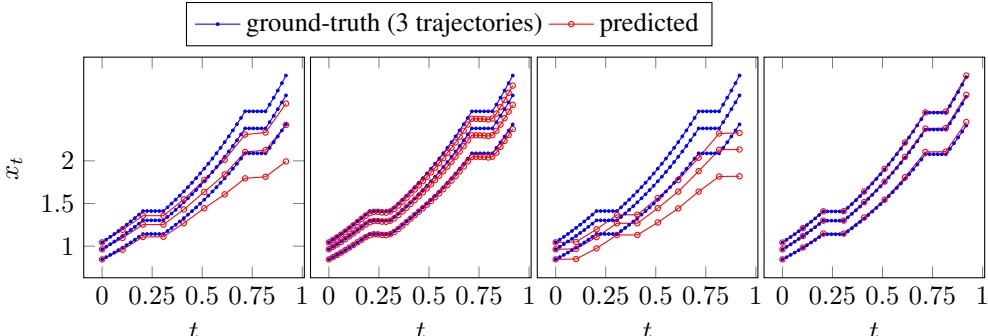

Figure 8: Comparisons between implicit and explicit losses and predictions for time-varying potentials for 3 different particles trajectories. From left to right: explicit loss (16) - explicit prediction (13), explicit loss (16) - explicit prediction (13) with more observations, implicit loss (17) - explicit prediction (13), implicit loss (17) - implicit prediction (15).

When only the potential energy is considered, one can adopt the implicit scheme $x_{t+1} = x_t - \tau \nabla V(x_{t+1})$ which is also suggested by the loss in Proposition 3.1. In particular, we adopt this scheme for the time-varying potential in Section 4.4:

$$x_{t+1} = x_t - \tau \nabla V(x_{t+1}, t+1). \tag{15}$$

We refer to this scheme as *implicit*, and to (13) as *explicit*, since in (15) $x_{t+1}$ appears in both sides of the equation and its value is found solving an optimization problem, whereas in (13) it can be computed directly.

**Implicit vs explicit prediction scheme with time-varying potential.** We now discuss one potential pitfall when training JKOnet$^*$ with time-varying potentials. For this, we work on $\mathbb{R}$ (instead of the space of probability measures) and consider the analysis in Section 3.1 for the time-varying function

$$V(x,t) = \begin{cases} 0 & \text{if } 0.2 \le t \le 0.3 \text{ or } 0.7 \le t \le 0.8, \\ -0.75 \cdot x^2 & \text{otherwise.} \end{cases}$$

Suppose one adapts the loss in (7) so that the value of the potential at the previous timestep is used, i.e.,

$$\sum_{t=0}^{T-1} \int_{\mathbb{R}^d \times \mathbb{R}^d} \left\| \boldsymbol{\nabla} V(x_{t+1}, t) + \frac{1}{\tau}(x_{t+1} - x_t) \right\|^2 \mathrm{d}\gamma_t(x_t, x_{t+1}). \tag{16}$$

This *explicit* adaptation, in our experiments, requires more observations to correctly recover the energy potential; see the first two figures on the left in Figure 8. Since in Section 4.4 we are given only 5 timesteps, we explored the use of an *implicit* adaptation, i.e.,

$$\sum_{t=0}^{T-1} \int_{\mathbb{R}^d \times \mathbb{R}^d} \left\| \boldsymbol{\nabla} V(x_{t+1}, t+1) + \frac{1}{\tau}(x_{t+1} - x_t) \right\|^2 \mathrm{d}\gamma_t(x_t, x_{t+1}). \tag{17}$$

However, if the predictions are made with the explicit scheme in (13), the third figure in Figure 8 is obtained, which suggests that the predictions are shifted in time. With the implicit prediction scheme in (15), instead, we are able to recover the correct potential with only 10 timesteps; see the right-most figure in Figure 8. We rely on these considerations to learn the time-varying energy potential of the scRNA-seq data in Section 4.4.

## C  Training and model details

This section describes the training details for our model JKOnet$^*$.

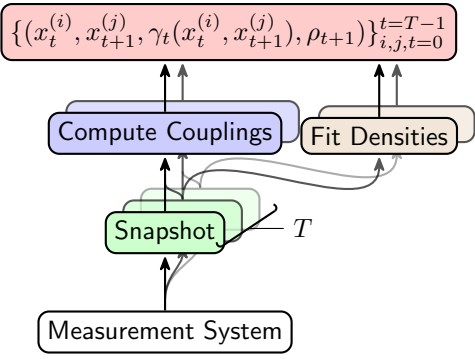

Figure 9: Data pipeline for JKOnet$^*$.

## C.1 Data pre-processing

In principle, a measurement system is used to take a trajectory of populations. Then, we compute the optimal couplings between consecutive snapshots and, if needed, infer the density of each snapshot. This information is then combined to get the training set (see Figure 9).

## C.2 Computation of the optimal couplings

The raw population data $\mu_0, \mu_1, \ldots, \mu_T$ are typically empirical. Since they are not optimization variables in the learning task, the optimal couplings $\gamma_0, \gamma_1, \ldots, \gamma_T$ can be computed beforehand. Differently from [9], we do not need to track the gradients within the Sinkhorn algorithm to compute the back-propagation [13], and we do not need to solve any optimal transport problem during training. If the number of particles per snapshot is very large, one may consider splitting the single snapshot into multiple ones and computing the couplings of the sub-sampled population. For datasets larger than 1000 particles, we compute the couplings in batches of 1000. In our experiments, we solve the optimal transport problem via linear programming using the POT library [18]. In Figure 10, we compare the performances resulting from training JKOnet$^*$ to recover the potentials in Appendix F when the couplings are computed with plain linear programming or with Sinkhorn-type algorithms [13] with various degrees of regularization $\varepsilon$. In particular, we conclude that, as long as the couplings are close to the correct one, the algorithm used to compute them does not impact the performance of JKOnet$^*$. However, small regularizers slow down the Sinkhorn algorithm and, thus, we prefer to directly solve the linear program without regularization. In general, the solver choice can be considered an additional knob that researchers and practitioners can tune when deploying JKOnet$^*$.

To instead investigate how the dimensionality affects the computation of the optimal transport plan, recall that the optimal transport problem between two empirical measures $\mu_1 = \sum_{i=1}^{N} \mu(x_i)\delta_{x_i}$ and $\mu_2 = \sum_{j=1}^{M} \mu(y_j)\delta_{y_j}$ can be cast as the linear program

$$\min_{\gamma_{ij} \geq 0} \sum_{ij} c_{ij}\gamma_{ij}$$

$$\text{s.t.} \sum_{i=1}^{N} \gamma_{ij} = \mu_2(y_j) \quad \forall j \in \{1, \ldots, M\}$$

$$\sum_{j=1}^{M} \gamma_{ij} = \mu_1(x_i) \quad \forall i \in \{1, \ldots, N\},$$

where $\gamma_{ij}$ is the weight of the coupling between $x_i$ and $y_j$, $\gamma(x_i, y_j) = \gamma_{ij}$ and $c_{ij} = \|x_i - x_j\|^2$ is the $ij^{\text{th}}$ entry of the cost matrix $C$. Therefore, the dimensionality $d$ of the data impacts the solving time only in relation to the construction of the cost matrix $C$ (in particular, linearly). However, this cost is negligible and dwarfed by the actual solution of the linear program. In particular, the dimensionality does not affect the size of the linear program. Moreover, the optimal plans at different timesteps can be computed in parallel. As a result, only the number of particles (i.e., the size of the

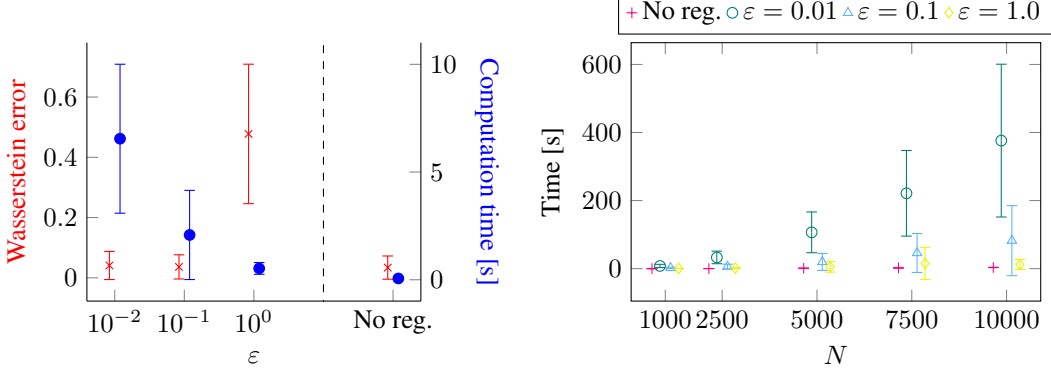

Figure 10: Comparison of the performances resulting from training JKOnet* to recover the potentials when the couplings are computed with plain linear programming (No reg.) or with Sinkhorn-type algorithms with regularization $\varepsilon$. In the plot on the left, the left y-axis (in red, crosses) represents the EMD error (normalized so that 0 and 1 are the minimum and maximum obtained with the different methods), and the right $y$-axis (in blue, circles) represents the computation time in seconds. In the plot on the left, we report the computational time for the different methods for different numbers of particles. The statistics are computed on all the data generated with the potentials listed in Appendix F with the same settings as in Sections 4.1 and 4.2, with no batching for the optimal transport couplings computation and using the default configuration in the POT library [18] for linear programming and in the OTT-JAX library [6] for the Sinkhorn algorithm.

dataset) is a scaling bottleneck. In practice, as showcased by the real-world application in Section 4.4, this is often not an issue. For completeness, we report an ablation on the number of particles for the computation times of plain linear programming and Sinkhorn algorithm with different values of the regularizer in Figure 10, using the default configuration in the POT library [18] for linear programming and in the OTT-JAX library [6] for the Sinkhorn algorithm. This analysis suggests that, while the number of particles affects the computational time in an exponential fashion, the computation of optimal transport plans does not pose a problem for reasonably sized datasets. Also, we highlight that this operation is done only once and independently from, e.g., the network architecture. Finally, as discussed above, batching strategies can help reduce the computational cost. Specifically, rather than computing the couplings between $N$ particles, one can compute $\lceil N/b \rceil$ couplings between $b$ particles, sampled uniformly from the entire population at each timestep, and then aggregate them.

### C.3 Estimation of $\rho$ and $\nabla \rho$

The case with $\theta_3$ non-zero imposes the estimation of the density $\rho_t$ and its gradient $\nabla \rho_t$ from the empirical probability measures $\mu_t$. To the extent of estimating $\rho_t$, there are many viable options (e.g., see the SciPy package [53]). We use a mixture of 10 gaussians for all the experiments in this paper. To compute $\nabla \rho_t$, we rely on the autograd feature of JAX [6]. The complexity related to $\rho_t$ and $\nabla \rho_t$, instead, is bypassed if there is no internal energy in (10) (i.e., $\theta_3 = 0$).

### C.4 Optimizer

We use the Adam optimizer [26] with the parameters $\beta_1 = 0.9, \beta_2 = 0.999, \varepsilon = 1\text{e-8}$, and constant learning rate lr = 1e-3. The model is trained with gradient clipping with maximum global norm for an update of 10. We process the data in batches of 250.

### C.5 Network architecture

The neural networks of potential and interaction energies are multi-layer perceptrons with 2 hidden layers of size 64 with softplus activation functions and a one-dimensional output layer (cf. [9]). Future work shall investigate different architectures for the interaction energy, for instance using (Set-)Transformers [31, 52] or Deep Sets [55].

## C.6 Features selection for the linear approximation

The features used in this work are polynomials up to degree $4$ and radial basis exponential functions $\exp\left(-\frac{\|v-c\|^2}{\sigma}\right)$ with $\sigma = 0.5$ and $c$ in the discretized (10 points per dimension) grid $[-4,4]^d$. Finally, we use a regularizer $\lambda = 0.01$ on the square norm of the parameters $\theta = \left[\theta_1^\top, \theta_2^\top, \theta_3^\top\right]^\top$.

## C.7 Hardware

The empirical data was collected entirely on an Ubuntu 22.04 machine equipped with an AMD Ryzen Threadripper PRO 5995WX processor and a Nvidia RTX 4090 GPU. Interestingly, both `JKOnet`$^*$ and `JKOnet`$^*_V$ demonstrate comparable computational times when executed on a GPU, highlighting the method's strong parallelizability. Among the methods compared in Section 4.1, `JKOnet`$^*$ is the only one that significantly benefits from GPU parallelization in our experiments, while all others exhibit similar computational times when run on a CPU. When parallelizing over CPU cores, we used on `parallel` [46].

# D Proofs

## D.1 Preliminaries

We briefly collect and summarizes the notation, definitions, and results of [30] used in the proofs of this work. For more intuition and details, we refer the reader to [30]. A sequence $(\mu_n)_{n\in\mathbb{N}} \subseteq \mathcal{P}(\mathbb{R}^d)$ converges narrowly to $\mu$ if $\int_{\mathbb{R}^d} \phi(x)\mathrm{d}\mu_n(x) \to \int_{\mathbb{R}^d} \phi(x)\mathrm{d}\mu(x)$ for all bounded continuous functions $\phi : \mathbb{R}^d \to \mathbb{R}$. We say that the convergence is "in Wasserstein" if $W_2(\mu_n, \mu) \to 0$ or, equivalently, $\int_{\mathbb{R}^d} \phi(x)\mathrm{d}\mu_n(x) \to \int_{\mathbb{R}^d} \phi(x)\mathrm{d}\mu(x)$ for all continuous functions $\phi : \mathbb{R}^d \to \mathbb{R}$ with at most quadratic growth (i.e., $\phi(x) \leq A(1 + \|x\|^2)$ for some $A > 0$). For $i, m \in \mathbb{N}, 1 \leq i \leq m$, we denote by $\pi_i : (\mathbb{R}^d)^m \to \mathbb{R}^d$ the projection map on the $i^{\text{th}}$ component, i.e., $\pi_i(x_1, \ldots, x_m) = x_i$. We compose projections as $\pi_{ij}(x) = (\pi_i(x), \pi_j(x))$.

**Wasserstein calculus.** We use two notions of subdifferentability in the Wasserstein space.

- *Regular Wasserstein subdifferentiability.* A transport plan $\bar{\xi} \in \mathcal{P}(\mathbb{R}^d \times \mathbb{R}^d)$ belongs to the regular subdifferential of a functional $J : \mathcal{P}(\mathbb{R}^d) \to \bar{\mathbb{R}}$ at $\bar{\mu} \in \mathcal{P}(\mathbb{R}^d)$, denoted $\hat{\partial}J(\bar{\mu})$, if for all $\mu \in \mathcal{P}(\mathbb{R}^d)$ and $\xi \in (\pi_1, \pi_2 - \pi_1)_{\#}\Gamma_0(\bar{\mu}, \mu)$ we have

$$J(\mu) - J(\bar{\mu}) \geq \max_{\substack{\alpha \in \mathcal{P}(\mathbb{R}^d \times \mathbb{R}^d \times \mathbb{R}^d) \\ \pi_{12\#}\alpha = \bar{\xi}, \pi_{13\#}\alpha = \xi}} \int_{\mathbb{R}^d \times \mathbb{R}^d \times \mathbb{R}^d} \bar{v}^\top v \, \mathrm{d}\alpha(\bar{x}, \bar{v}, v) + o(W_2(\bar{\mu}, \mu)).$$

  Here, $\alpha$ can be interpreted as a coupling of the tangent vectors $\bar{\xi}$ (the subgradient) and $\xi$ (the tangent vector $\mu - \bar{\mu}$).

- *(General) Wasserstein subdifferentiability.* A transport plan $\bar{\xi} \in \mathcal{P}(\mathbb{R}^d \times \mathbb{R}^d)$ belongs to the (general) subdifferential of a functional $J : \mathcal{P}(\mathbb{R}^d) \to \bar{\mathbb{R}}$ at $\bar{\mu}$, denoted $\partial J(\bar{\mu})$, if there are sequences $(\mu_n)_{n\in\mathbb{N}} \subset \mathcal{P}(\mathbb{R}^d)$ and $(\xi_n)_{n\in\mathbb{N}} \subset \mathcal{P}(\mathbb{R}^d \times \mathbb{R}^d)$ so that (i) $\xi_n \in \hat{\partial}J(\mu_n)$, (ii) $\xi_n$ converges in Wasserstein to $\bar{\xi}$.

From these definitions, the analogous ones for the supergradient follows: The regular supergradients of $J$ at $\bar{\mu}$ are the elements of $-\hat{\partial}(-J)(\bar{\mu})$ and the supergradients as the elements of $-\partial(-J)(\bar{\mu})$. We then call the *gradient* of $J$ the unique element, if it exists, of $-\hat{\partial}(-J)(\bar{\mu}) \cap \hat{\partial}J(\bar{\mu})$, and we say that $J$ is differentiable.

**Necessary Conditions for Optimality.** For $J$ proper and lower semi-continuous w.r.t. convergence in Wasserstein (i.e., if $\mu_n$ converges in Wasserstein to $\mu$, then $\liminf_{n\to\infty} J(\mu_n) \geq J(\mu)$), consider the optimization problem

$$\inf_{\mu \in \mathcal{P}(\mathbb{R}^d)} J(\mu). \tag{18}$$

Then, if $\mu^* \in \mathcal{P}(\mathbb{R}^d)$ is optimal for (18), i.e., $J(\mu^*) = \inf_{\mu \in \mathcal{P}(\mathbb{R}^d)} J(\mu)$, it must satisfy the "Fermat's rule in Wasserstein space" (see [30, Theorem 3.3]):

$$0_{\mu^*} := \mu^* \times \delta_0 \in \partial J(\mu^*). \tag{19}$$

In particular, this implies that for some $\xi \in \partial J(\mu^*)$ it must hold

$$0 = \int_{\mathbb{R}^d \times \mathbb{R}^d} \|v\|^2 \mathrm{d}\xi(x^*, v). \tag{20}$$

**Outline of the appendix and proofs strategy** Although the statement of Proposition 3.1 follows directly from Proposition 3.2, we prove it separately in Appendix D.3 as it is the simplest setting and, thus, the easiest for the reader to familiarize with the techniques used in this work. Since the internal energy (perhaps surprisingly) simplifies the setting by restricting to absolutely continuous measures (i.e., $\{\mu \in \mathcal{P}(\mathbb{R}^d) : \mu \ll \mathrm{d}\mathcal{L}\}$), we prove also the statement for potential and interaction energy only in Appendix D.4. We then provide the proof for the most general statement in Appendix D.5. All the proofs follow the same recipe. First, we prove existence of a solution. Second, we characterize the Wasserstein subgradients for the functional considered. Finally, we conclude deploying [30, Theorem 3.3]. Because the Wasserstein subgradients will be constructed using the subdifferential calculus rules (cf. [30, Proposition 2.17 and Corollary 2.18]), we collect them all in Appendix D.2. This appendix ends with Appendix D.6, in which we prove Proposition 3.4 using standard $\mathbb{R}^d$ optimization arguments.

## D.2  Wasserstein subgradients

**(Scaled) Wasserstein distance.** By [30, Corollary 2.12], the (Wasserstein) subgradients of $\frac{1}{2}W_2(\mu, \mu_t)^2$ at $\mu \in \mathcal{P}(\mathbb{R}^d)$ are all of the form $(\pi_1, \pi_1 - \pi_2)_\# \gamma_t$, for an optimal transport plan $\gamma_t$ between $\mu$ and $\mu_t$. Then, by [30, Corollary 2.18], the subgradients of $\frac{1}{2\tau}W_2(\mu, \mu_t)^2$ are

$$\left(\pi_1, \frac{\pi_1 - \pi_2}{\tau}\right)_\# \gamma_t. \tag{21}$$

Since the Wasserstein distance is in general not regularly subdifferentiable (cf. [30, Proposition 2.6]), it is not differentiable.

**Potential energy.** Under the assumptions in Propositions 3.1, 3.2 and 3.4, we deploy [30, Proposition 2.13] to conclude that the potential energy is differentiable at $\mu \in \mathcal{P}(\mathbb{R}^d)$ with gradient given by

$$(\mathrm{Id}, \boldsymbol{\nabla} V)_\# \mu. \tag{22}$$

**Interaction energy.** Under the assumptions in Propositions 3.2 and 3.4, we deploy [30, Proposition 2.15] to conclude that the interaction energy is differentiable at $\mu \in \mathcal{P}(\mathbb{R}^d)$ with gradient given by

$$\left(\mathrm{Id}, \int_{\mathbb{R}^d} \boldsymbol{\nabla} U(\mathrm{Id} - x) \mathrm{d}\mu(x)\right)_\# \mu. \tag{23}$$

**Internal energy.** Under the assumptions in Propositions 3.2 and 3.4, we deploy [29, Example 2.3] and the consistency of the tangent space (cf. [30, §2.2]) to conclude that the internal energy is differentiable at $\mu \ll \mathrm{d}x$ with gradient given by

$$\left(\mathrm{Id}, \frac{\boldsymbol{\nabla}\rho}{\rho}\right)_\# \mu. \tag{24}$$

In particular, here we consider $\mu \ll \mathrm{d}x$ since the internal energy is otherwise $+\infty$ by definition and $\mu$ is certainly not a minimum.

## D.3  Proof of Proposition 3.1

The JKO step in Proposition 3.1 is the optimization problem, resembling (18),

$$\inf_{\mu \in \mathcal{P}(\mathbb{R}^d)} \left\{ J(\mu) := \int_{\mathbb{R}^d} V(x) \mathrm{d}\mu(x) + \frac{1}{2\tau} W_2(\mu, \mu_t)^2 \right\}.$$

Since $V$ is lower bounded, up to replacing $V$ by $V - \min_{x \in \mathbb{R}^d} V(x)$, we can without loss of generality assume that $V$ is non-negative. We now proceed in three steps.

**Existence of a solution.**

- *J has closed level sets w.r.t. narrow convergence.* As an intermediate step to prove compactness of the level sets we prove their closedness, which is equivalent to lower semi-continuity of $J$. Since $V$ is continuous and lower bounded, the functional $\mu \mapsto \int_{\mathbb{R}^d} V(x)\mathrm{d}\mu(x)$ is lower semi-continuous [29]. The Wasserstein distance is well-known to be lower semi-continuous [30]. Thus, their sum $J$ is also lower semi-continuous.

- *J has compact (w.r.t. narrow convergence) level sets.* Without loss generality, assume $\lambda$ is large enough so that the level set is not empty. Otherwise, the statement is trivial. Closedness of the level sets follows directly from lower semi-continuity, thus it suffices to prove that level sets are contained in a compact set, because closed subsets of compact spaces are compact. By non-negativity of $V$,

$$\{\mu \in \mathcal{P}(\mathbb{R}^d) : J(\mu) \leq \lambda\} \subset \left\{\mu \in \mathcal{P}(\mathbb{R}^d) : \frac{1}{2\tau}W_2(\mu, \mu_t)^2 \leq \lambda\right\}$$
$$= \{\mu \in \mathcal{P}(\mathbb{R}^d) : W_2(\mu, \mu_t)^2 \leq 2\tau\lambda\}.$$

Since the Wasserstein distance has compact level sets [29] w.r.t. narrow convergence, we conclude.

- *Compact level sets imply existence of a solution.* Let $\alpha := \inf_{\mu \in \mathcal{P}(\mathbb{R}^d)} J(\mu)$ and let $(\mu_n)_{n \in \mathbb{N}}$ be a minimizing sequence so that $J(\mu_n) \to \alpha$. Then, by definition of convergence, $\mu_n \in \{\mu \in \mathcal{P}(\mathbb{R}^d) : J(\mu) \leq 2\alpha\}$ for all $n$ sufficiently large. By compactness of the level sets, $\mu_n$ converges narrowly (up to subsequences) to some $\mu \in \{\mu \in \mathcal{P}(\mathbb{R}^d) : J(\mu) \leq \lambda\}$. Since compactness of the level sets implies closedness and a functional with closed level sets is lower semi-continuous (w.r.t. narrow convergence) we conclude that $J(\mu) \leq \liminf_{n \to \infty} J(\mu_n) = \alpha$. Thus, $\mu$ is a minimizer of $J$.

**Wasserstein subgradients.** By [30, Proposition 2.17], we combine (21) and (22) to conclude that any subgradient of $J$ at $\mu_{t+1}$ must be of the form $(\pi_1, \pi_2 + \pi_3)_{\#}\alpha$, for some "sum" coupling $\alpha \in \mathcal{P}(\mathbb{R}^d \times \mathbb{R}^d \times \mathbb{R}^d)$, $\pi_{12\#}\alpha = (\mathrm{Id}, \boldsymbol{\nabla}V)_{\#}\mu_{t+1}$ and $\pi_{13\#}\alpha = \left(\pi_1, \frac{\pi_1 - \pi_2}{\tau}\right)_{\#}\gamma'_{t+1}$ (cf. [30, Definition 2.1]), with $\gamma'_{t+1}$ being an optimal transport plan between $\mu_{t+1}$ and $\mu_t$.

**Necessary condition for optimality.** Any minimizer $\mu_{t+1}$ of $J$ satisfies $0_{\mu_{t+1}} \in \partial J(\mu_{t+1})$. Thus, for $\mu_{t+1}$ to be optimal, there must exist $\gamma'_{t+1}$ and $\alpha$, so that (cf. (20))

$$0 = \int_{\mathbb{R}^d \times \mathbb{R}^d \times \mathbb{R}^d} \|v_2 + v_3\|^2 \mathrm{d}\alpha(x_{t+1}, v_1, v_2)$$
$$= \int_{\mathbb{R}^d \times \mathbb{R}^d \times \mathbb{R}^d} \|\boldsymbol{\nabla}V(x_{t+1}) + v_2\|^2 \mathrm{d}\alpha(x_{t+1}, v_1, v_2)$$
$$= \int_{\mathbb{R}^d \times \mathbb{R}^d} \|\boldsymbol{\nabla}V(x_{t+1}) + v_2\|^2 \mathrm{d}((\pi_{13})_{\#}\alpha)(x_{t+1}, v_2)$$
$$= \int_{\mathbb{R}^d \times \mathbb{R}^d} \|\boldsymbol{\nabla}V(x_{t+1}) + v_2\|^2 \mathrm{d}\left(\left(\pi_1, \frac{\pi_1 - \pi_2}{\tau}\right)_{\#}\gamma'_{t+1}\right)(x_{t+1}, v_2)$$
$$= \int_{\mathbb{R}^d \times \mathbb{R}^d} \left\|\boldsymbol{\nabla}V(x_{t+1}) + \frac{x_{t+1} - x_t}{\tau}\right\|^2 \mathrm{d}\gamma'_{t+1}(x_{t+1}, x_t).$$

Finally, consider $\gamma_t = (\pi_2, \pi_1)_{\#}\gamma'_{t+1}$ to conclude.

## D.4 The case of potential and interaction energies

As a preliminary step to Proposition 3.2, consider the case $\beta = 0$. The JKO step is the optimization problem, resembling (18),

$$J(\mu) := \int_{\mathbb{R}^d} V(x)\mathrm{d}\mu(x) + \int_{\mathbb{R}^d \times \mathbb{R}^d} U(x - y)\mathrm{d}(\mu \times \mu)(x, y) + \frac{1}{2\tau}W_2(\mu, \mu_t)^2. \quad (25)$$

Since $V$ and $U$ are lower bounded, up to replacing $J$ with $J - \inf_{\mu \in \mathcal{P}(\mathbb{R}^d)} J(\mu)$, we can without loss of generality assume that $J$ is non-negative.

**Existence of a solution.** Since also $\mu \mapsto \int_{\mathbb{R}^d \times \mathbb{R}^d} U(x-y)\mathrm{d}(\mu \times \mu)(x,y)$ is lower semi-continuous w.r.t. narrow convergence [29], the proof of existence is analogous to the one in Proposition 3.1.

**Wasserstein subgradients.** By [30, Proposition 2.17], we combine (22) and (23) to conclude that each subgradient of the functional

$$\mu \mapsto \int_{\mathbb{R}^d} V(x)\mathrm{d}\mu(x) + \int_{\mathbb{R}^d} \int_{\mathbb{R}^d} U(x-y)\mathrm{d}\mu(y)\mathrm{d}\mu(x) \tag{26}$$

at $\mu_{t+1}$ is of the form $(\pi_1, \pi_2 + \pi_3)_{\#}\alpha'$ for some coupling $\alpha' \in \mathcal{P}(\mathbb{R}^d \times \mathbb{R}^d \times \mathbb{R}^d)$ such that $\pi_{12\#}\alpha' = (\mathrm{Id}, \boldsymbol{\nabla} V)_{\#}\mu_{t+1}$ and $\pi_{13\#}\alpha' = \left(\mathrm{Id}, \int_{\mathbb{R}^d} \boldsymbol{\nabla} U(\mathrm{Id}-x)\mathrm{d}\mu_{t+1}(x)\right)_{\#}\mu_{t+1}$. But then $\alpha' = \left(\mathrm{Id}, \boldsymbol{\nabla} V, \int_{\mathbb{R}^d} \boldsymbol{\nabla} U(\mathrm{Id}-x)\mathrm{d}\mu_{t+1}(x)\right)_{\#}\mu_{t+1}$ and (26) has the unique subgradient

$$\left(\mathrm{Id}, \boldsymbol{\nabla} V + \int_{\mathbb{R}^d} \boldsymbol{\nabla} U(\mathrm{Id}-x)\mathrm{d}\mu_{t+1}(x)\right)_{\#}\mu_{t+1}. \tag{27}$$

Similarly, we conclude that $\left(\mathrm{Id}, -\left(\boldsymbol{\nabla} V + \int_{\mathbb{R}^d} \boldsymbol{\nabla} U(\mathrm{Id}-x))\mathrm{d}\mu(x)\right)\right)_{\#}\mu_{t+1}$ is the unique supergradient of (26) at $\mu$ and, thus, it is differentiable there. We can thus deploy again [30, Proposition 2.17] to combine (27) with (21) and conclude that the subgradients of (25) are of the form $(\pi_1, \pi_2 + \pi_3)_{\#}\alpha$, for $\alpha \in \mathcal{P}(\mathbb{R}^d \times \mathbb{R}^d \times \mathbb{R}^d)$, $\pi_{12\#}\alpha = \left(\mathrm{Id}, \boldsymbol{\nabla} V + \int_{\mathbb{R}^d} \boldsymbol{\nabla} U(\mathrm{Id}-x)\mathrm{d}\mu_{t+1}(x)\right)_{\#}\mu_{t+1}$ and $(\pi_1, \pi_2)_{\#}\alpha = \left(\pi_1, \frac{\pi_1 - \pi_2}{\tau}\right)_{\#}\gamma'_{t+1}$ (cf. [30, Definition 2.1]), with $\gamma'_{t+1}$ being an optimal transport plan between $\mu_{t+1}$ and $\mu_t$.

**Necessary condition for optimality.** The steps are analogous to Appendix D.3. Any minimizer $\mu_{t+1}$ of $J$ satisfies $0_{\mu_{t+1}} \in \partial J(\mu_{t+1})$. Thus, for $\mu_{t+1}$ to be optimal, there must exist $\gamma'_{t+1}$ and $\alpha$, so that (cf. (20))

$$0 = \int_{\mathbb{R}^d \times \mathbb{R}^d \times \mathbb{R}^d} \|v_2 + v_3\|^2 \mathrm{d}\alpha(x_{t+1}, v_1, v_2)$$

$$= \int_{\mathbb{R}^d \times \mathbb{R}^d \times \mathbb{R}^d} \left\|\boldsymbol{\nabla} V(x_{t+1}) + \int_{\mathbb{R}^d} \boldsymbol{\nabla} U(x_{t+1} - x)\mathrm{d}\mu_{t+1}(x) + v_2\right\|^2 \mathrm{d}\alpha(x_{t+1}, v_1, v_2)$$

$$= \int_{\mathbb{R}^d \times \mathbb{R}^d} \left\|\boldsymbol{\nabla} V(x_{t+1}) + \int_{\mathbb{R}^d} \boldsymbol{\nabla} U(x_{t+1} - x)\mathrm{d}\mu_{t+1}(x) + v_2\right\|^2 \mathrm{d}((\pi_{13})_{\#}\alpha)(x_{t+1}, v_2)$$

$$= \int_{\mathbb{R}^d \times \mathbb{R}^d} \left\|\boldsymbol{\nabla} V(x_{t+1}) + \int_{\mathbb{R}^d} \boldsymbol{\nabla} U(x_{t+1} - x)\mathrm{d}\mu_{t+1}(x) + \frac{x_{t+1} - x_t}{\tau}\right\|^2 \mathrm{d}\gamma'_{t+1}(x_{t+1}, x_t).$$

Finally, consider $\gamma_t = (\pi_2, \pi_1)_{\#}\gamma'_{t+1}$ to conclude. Importantly, for $\beta = 0$ we do not need to restrict to $\mu \ll \mathrm{d}x$ in Proposition 3.2, so that the statement holds regardless of $\beta$ (see also Appendix D.5).

### D.5 Proof of Proposition 3.2

For $\beta = 0$, see Appendix D.4. Let $\beta > 0$ here. The JKO step in Proposition 3.1 is the optimization problem, resembling (18),

$$J(\mu) := \begin{cases} \int_{\mathbb{R}^d} V(x)\mathrm{d}\mu(x) + \int_{\mathbb{R}^d \times \mathbb{R}^d} U(x-y)\mathrm{d}(\mu \times \mu)(x,y) \\ \quad + \beta \int_{\mathbb{R}^d} \rho(x)\log(x)\mathrm{d}x + \frac{1}{2\tau}W_2(\mu, \mu_t)^2 & \text{if } \mu \ll \mathcal{L}, \\ +\infty & \text{else.} \end{cases} \tag{28}$$

Since $V, U$, and $\int_{\mathbb{R}^d} \rho(x)\log(x)\mathrm{d}x$ are lower bounded, up to replacing $J$ with $J - \inf_{\mu \in \mathcal{P}(\mathbb{R}^d)} J(\mu)$, we can without loss of generality assume that $J$ is non-negative.

**Existence of a solution.** Since also $\mu \mapsto \int_{\mathbb{R}^d \times \mathbb{R}^d} U(x-y)\mathrm{d}(\mu \times \mu)(x,y)$ and

$$\mu \mapsto \begin{cases} \int_{\mathbb{R}^d} \rho(x)\log(x)\mathrm{d}x & \text{if } \mu \ll \mathcal{L}, \\ +\infty & \text{else.} \end{cases}$$

are lower semi-continuous w.r.t. narrow convergence [29], the proof of existence is analogous to the one in Proposition 3.1.

**Wasserstein subgradients.** When $\beta > 0$, only $\mu_{t+1} \ll \mathrm{d}\mathcal{L}$ can be optimal and we are thus interested in characterizing the subgradients of (28) only at these probability measures. Let $\rho_{t+1}$ be the density of $\mu_{t+1}$. Consider first the functional

$$\mu_{t+1} \mapsto \int_{\mathbb{R}^d} V(x)\mathrm{d}\mu_{t+1}(x) + \int_{\mathbb{R}^d}\int_{\mathbb{R}^d} U(x-y)\mathrm{d}\mu_{t+1}(y)\mathrm{d}\mu_{t+1}(x) + \beta \int_{\mathbb{R}^d} \rho_{t+1}(x)\log(\rho_{t+1}(x))\mathrm{d}x.$$
(29)

By [30, Proposition 2.17], we combine (27) and (24) to conclude that each subgradient of the functional (29) at $\mu_{t+1}$ is of the form $(\pi_1, \pi_2 + \pi_3)_{\#}\alpha'$ for some coupling $\alpha' \in \mathcal{P}(\mathbb{R}^d \times \mathbb{R}^d \times \mathbb{R}^d)$ such that $\pi_{12\#}\alpha' = \left(\mathrm{Id}, \boldsymbol{\nabla}V + \int_{\mathbb{R}^d} \boldsymbol{\nabla}U(\mathrm{Id}-x)\mathrm{d}\mu_{t+1}(x)\right)_{\#}\mu_{t+1}$ and $\pi_{13\#}\alpha' = \left(\mathrm{Id}, \beta\frac{\boldsymbol{\nabla}\rho_{t+1}}{\rho_{t+1}}\right)_{\#}\mu_{t+1}$. But then $\alpha' = \left(\mathrm{Id}, \boldsymbol{\nabla}V + \int_{\mathbb{R}^d} \boldsymbol{\nabla}U(\mathrm{Id}-x)\mathrm{d}\mu_{t+1}(x), \beta\frac{\boldsymbol{\nabla}\rho_{t+1}}{\rho_{t+1}}\right)_{\#}\mu_{t+1}$ and (26) has the unique subgradient

$$\left(\mathrm{Id}, \boldsymbol{\nabla}V + \int_{\mathbb{R}^d} \boldsymbol{\nabla}U(\mathrm{Id}-x)\mathrm{d}\mu_{t+1}(x) + \beta\frac{\boldsymbol{\nabla}\rho_{t+1}}{\rho_{t+1}}\right)_{\#}\mu_{t+1}.$$
(30)

However, note that now (26) is not differentiable, as it can attain the value $+\infty$. We can thus deploy again [30, Proposition 2.17], together with the fact the squared Wasserstein distance is differentible at absolutely continuous measures [30, Proposition 2.6], to combine (30) with (21) and conclude that the subgradients of (25) are of the form $(\pi_1, \pi_2 + \pi_3)_{\#}\alpha$, for $\alpha \in \mathcal{P}(\mathbb{R}^d \times \mathbb{R}^d \times \mathbb{R}^d)$, $\pi_{12\#}\alpha = \left(\mathrm{Id}, \boldsymbol{\nabla}V + \int_{\mathbb{R}^d} \boldsymbol{\nabla}U(\mathrm{Id}-x)\mathrm{d}\mu_{t+1}(x) + \beta\frac{\boldsymbol{\nabla}\rho_{t+1}}{\rho_{t+1}}\right)_{\#}\mu_{t+1}$ and $(\pi_1, \pi_2)_{\#}\alpha = \left(\pi_1, \frac{\pi_1 - \pi_2}{\tau}\right)_{\#}\gamma'_{t+1}$ (cf. [30, Definition 2.1]), with $\gamma'_{t+1}$ being an optimal transport plan between $\mu_{t+1}$ and $\mu_t$.

**Necessary condition for optimality.** The steps are analogous to Appendices D.3 and D.4. Any minimizer $\mu_{t+1}$ of $J$ satisfies $0_{\mu_{t+1}} \in \partial J(\mu_{t+1})$. Thus, for $\mu_{t+1}$ to be optimal, there must exist $\gamma'_{t+1}$ and $\alpha$, so that (cf. (20))

$$\begin{aligned}
0 &= \int_{\mathbb{R}^d \times \mathbb{R}^d \times \mathbb{R}^d} \|v_2 + v_3\|^2 \mathrm{d}\alpha(x_{t+1}, v_1, v_2) \\
&= \int_{\mathbb{R}^d \times \mathbb{R}^d \times \mathbb{R}^d} \left\| \boldsymbol{\nabla}V(x_{t+1}) + \int_{\mathbb{R}^d} \boldsymbol{\nabla}U(x_{t+1} - x)\mathrm{d}\mu_{t+1}(x) \right. \\
&\qquad\qquad\qquad \left. + \beta\frac{\boldsymbol{\nabla}\rho_{t+1}(x_{t+1})}{\rho_t(x_{t+1})} + v_2 \right\|^2 \mathrm{d}\alpha(x_{t+1}, v_1, v_2) \\
&= \int_{\mathbb{R}^d \times \mathbb{R}^d} \left\| \boldsymbol{\nabla}V(x_{t+1}) + \int_{\mathbb{R}^d} \boldsymbol{\nabla}U(x_{t+1} - x)\mathrm{d}\mu_{t+1}(x) \right. \\
&\qquad\qquad\qquad \left. + \beta\frac{\boldsymbol{\nabla}\rho_{t+1}(x_{t+1})}{\rho_t(x_{t+1})} + v_2 \right\|^2 \mathrm{d}((\pi_{13})_{\#}\alpha)(x_{t+1}, v_2) \\
&= \int_{\mathbb{R}^d \times \mathbb{R}^d} \left\| \boldsymbol{\nabla}V(x_{t+1}) + \int_{\mathbb{R}^d} \boldsymbol{\nabla}U(x_{t+1} - x)\mathrm{d}\mu_{t+1}(x) \right. \\
&\qquad\qquad\qquad \left. + \beta\frac{\boldsymbol{\nabla}\rho_{t+1}(x_{t+1})}{\rho_t(x_{t+1})} + \frac{x_{t+1} - x_t}{\tau} \right\|^2 \mathrm{d}\gamma'_{t+1}(x_{t+1}, x_t).
\end{aligned}$$

Finally, consider $\gamma_t = (\pi_2, \pi_1)_{\#}\gamma'_{t+1}$ to conclude.

## D.6 Proof of Proposition 3.4

We start with an explicit expression for the loss (11) in the case of linearly parametrized functionals:

$$\mathcal{L}(\theta_1, \theta_2, \theta_3) := \frac{1}{2} \sum_{t=0}^{T-1} \int_{\mathbb{R}^d \times \mathbb{R}^d} \left\| \boldsymbol{\nabla}\phi_1(x_{t+1})^\top \theta_1 + \left( \int_{\mathbb{R}^d} \boldsymbol{\nabla}\phi_2(x_{t+1} - x'_{t+1})^\top \mathrm{d}\mu_{t+1}(x'_{t+1}) \right) \theta_2 \right.$$
$$\left. + \theta_3 \frac{\boldsymbol{\nabla}\rho_{t+1}(x_{t+1})}{\rho_{t+1}(x_{t+1})} + \frac{1}{\tau}(x_{t+1} - x_t) \right\|^2 \mathrm{d}\gamma_t(x_t, x_{t+1}).$$

Since $J$ is convex and quadratic in $\theta$, we can find its minimum by setting the gradient to 0. In particular, the derivative of $\mathcal{L}$ w.r.t. $\theta_1$, $\boldsymbol{\nabla}_{\theta_1}\mathcal{L}(\theta_1, \theta_2, \theta_3)$, reads

$$\sum_{t=0}^{T-1} \int_{\mathbb{R}^d \times \mathbb{R}^d} \boldsymbol{\nabla}\phi_1(x_{t+1}) \left( \boldsymbol{\nabla}\phi_1(x_{t+1})^\top \theta_1 + \left( \int_{\mathbb{R}^d} \boldsymbol{\nabla}\phi_2(x_{t+1} - x'_{t+1})^\top \mathrm{d}\mu_{t+1}(x'_{t+1}) \right) \theta_2 \right.$$
$$\left. + \theta_3 \frac{\boldsymbol{\nabla}\rho_{t+1}(x_{t+1})}{\rho_{t+1}(x_{t+1})} + \frac{1}{\tau}(x_{t+1} - x_t) \right) \mathrm{d}\gamma_t(x_t, x_{t+1}).$$
$$= \sum_{t=0}^{T-1} \int_{\mathbb{R}^d \times \mathbb{R}^d} \boldsymbol{\nabla}\phi_1(x_{t+1}) \left( y_{t+1}(x_{t+1})^\top \theta + \frac{1}{\tau}(x_{t+1} - x_t) \right) \mathrm{d}\gamma_t(x_t, x_{t+1}),$$

where we used Leibniz integral rule to interchange gradient and integral. Similarly,

$$\boldsymbol{\nabla}_{\theta_2}\mathcal{L}(\theta) = \sum_{t=0}^{T-1} \int_{\mathbb{R}^d \times \mathbb{R}^d} \left( \int_{\mathbb{R}^d} \boldsymbol{\nabla}\phi_2(x_{t+1} - x'_{t+1})\mathrm{d}\mu_{t+1}(x'_{t+1}) \right)$$
$$\cdot \left( y_{t+1}(x_{t+1})^\top \theta + \frac{1}{\tau}(x_{t+1} - x_t) \right) \mathrm{d}\gamma_t(x_t, x_{t+1})$$

$$\boldsymbol{\nabla}_{\theta_3}\mathcal{L}(\theta) = \sum_{t=0}^{T-1} \int_{\mathbb{R}^d \times \mathbb{R}^d} \frac{\boldsymbol{\nabla}\rho_{t+1}(x_{t+1})^\top}{\rho_{t+1}(x_{t+1})} \left( y_{t+1}(x_{t+1})^\top \theta + \frac{1}{\tau}(x_{t+1} - x_t) \right) \mathrm{d}\gamma_t(x_t, x_{t+1}).$$

We split integrals and sums to get

$$\boldsymbol{\nabla}_{\theta_1}\mathcal{L}(\theta) = \left( \sum_{t=0}^{T-1} \int_{\mathbb{R}^d} \boldsymbol{\nabla}\phi_1(x_{t+1}) y_t(x_{t+1})^\top \mathrm{d}\mu_{t+1}(x_{t+1}) \right) \theta$$
$$+ \left( \sum_{t=0}^{T-1} \int_{\mathbb{R}^d \times \mathbb{R}^d} \frac{1}{\tau} \boldsymbol{\nabla}\phi_1(x_{t+1})(x_{t+1} - x_t)\mathrm{d}\gamma_t(x_t, x_{t+1}) \right)$$

$$\boldsymbol{\nabla}_{\theta_2}\mathcal{L}(\theta) = \left( \sum_{t=0}^{T-1} \int_{\mathbb{R}^d} \left( \int_{\mathbb{R}^d} \boldsymbol{\nabla}\phi_2(x_{t+1} - x'_{t+1})\mathrm{d}\mu_{t+1}(x'_{t+1}) \right) y_t(x_{t+1})^\top \mathrm{d}\mu_{t+1}(x_{t+1}) \right) \theta$$
$$+ \left( \sum_{t=0}^{T-1} \int_{\mathbb{R}^d \times \mathbb{R}^d} \frac{1}{\tau} \left( \int_{\mathbb{R}^d} \boldsymbol{\nabla}\phi_2(x_{t+1} - x'_{t+1})\mathrm{d}\mu_{t+1}(x'_{t+1}) \right) (x_{t+1} - x_t)\mathrm{d}\gamma_t(x_t, x_{t+1}) \right)$$

$$\boldsymbol{\nabla}_{\theta_3}\mathcal{L}(\theta) = \left( \sum_{t=0}^{T-1} \int_{\mathbb{R}^d} \frac{\boldsymbol{\nabla}\rho_{t+1}(x_{t+1})^\top}{\rho_{t+1}(x_{t+1})} y_t(x_{t+1})^\top \mathrm{d}\mu_{t+1}(x_{t+1}) \right) \theta$$
$$+ \left( \sum_{t=0}^{T-1} \int_{\mathbb{R}^d \times \mathbb{R}^d} \frac{1}{\tau} \frac{\boldsymbol{\nabla}\rho_{t+1}(x_{t+1})^\top}{\rho_{t+1}(x_{t+1})}(x_{t+1} - x_t)\mathrm{d}\gamma_t(x_t, x_{t+1}) \right).$$

We stack these expressions to compactly write the gradient of $\mathcal{L}$

$$\boldsymbol{\nabla}_{\theta}\mathcal{L}(\theta) = \left( \sum_{t=0}^{T-1} \int_{\mathbb{R}^d} y_t(x_{t+1}) y_t(x_{t+1})^\top \mathrm{d}\mu_{t+1}(x_{t+1}) \right) \theta$$
$$+ \frac{1}{\tau} \sum_{t=0}^{T-1} \int_{\mathbb{R}^d \times \mathbb{R}^d} y_t(x_{t+1})(x_{t+1} - x_t)\mathrm{d}\gamma_t(x_t, x_{t+1}).$$

The expression for $\theta$ follows solving $\nabla_\theta \mathcal{L}(\theta) = 0$ (and shifting the indices in the first sums from $t$ to $t'$ with $t' = t + 1$).

# E    Baselines settings and further comparisons

`JKOnet`.    We use the default configuration provided in [9]. Specifically:

- The potential energy is parametrized and optimized in the same way as for `JKOnet`$^*$ (cf. Appendices C.4 and C.5).

- The optimal transport map is parametrized with an ICNN with two layers of 64 neurons each, with positive weights and Gaussian initialization. The optimization of the transport map is performed with the Adam optimizer [26] with learning rate $\text{lr} = 0.01$, $\beta_1 = 0.5$, $\beta_2 = 0.9$ and $\varepsilon = 1e - 8$. The transport map is optimized with 100 iterations (i.e., we do not use the fixed point approach).

- We train the model for 100 epochs in batch sizes (number of particles per snapshot) of 250, without teacher forcing and without any convex regularization. The Sinkhorn algorithm was run with $\varepsilon = 1$.

`JKOnet` **vanilla.**    The only difference between this method and `JKOnet` is that we replace the ICNN with a vanilla MLP with two layers of 64 neurons each as parametrization of the optimal transport map.

`JKOnet` **with Monge Gap.**    The settings are the same as the vanilla `JKOnet`, with the Monge gap [51] as an additional term in the loss function for the optimal transport map (inner loop). For the computation of the Monge gap we used the recommended implementation in [51], namely the one in the `OTT-JAX` package [13]. The coefficient of the Monge gap regularization was set to $1.0$. No results are reported on the performance of this method because the computational effort experienced rendered the results not interesting a priori (they were on par with `JKOnet` but the time required per experiment is orders of magnitude more).

**Single-cell diffusion dynamics.**    For the methods in Section 4.4, we report the performances directly from [49, 12]. As mentioned in the main body of the paper, the performance of some of the methods are not one-to-one comparable with ours, due to a slightly different experimental setting. We further clarify the differences here. In our setting and the one used for `JKOnet`, we use 60% of the data at all time steps for training and 40% for testing. To evaluate performance, we compute the EMD between one-step-ahead predictions and ground truth, averaged over the testing data. In the setting of the other algorithms in Section 4.4, instead, all data from all time steps but one is used for training, and the data from the left-out time step is used for testing. Formally, if $t$ is the time step used for testing, then $\mu_t$ is used for testing and $\mu_k, k \neq t$ is used for training. Note that, in all algorithms, the first and last time steps are always used for training. Since the data includes 5 time steps and one is left out, 80% of the data is used for training and 20% for testing (in practice, these numbers are only approximate since the number of particles is not constant over time). In this case, performance is evaluated by computing EMD between the prediction and the ground truth at the time step left out, averaged over the left-out time steps. The differences between the two settings demand some caution in comparing and interpreting the performance of the various algorithms. Accordingly, we limit ourselves to observing that `JKOnet`$^*$ achieves state-of-the-art performance while requiring significantly lower training time (a few minutes compared to hours of the other methods).

# F    Functionals

We tested our methods on a selection of functionals from standard optimization tests [45]. We report the functionals used with (a projection of) their level set (see Figure 2) for completeness and reproducibility. For $v \in \mathbb{R}^d$, we conveniently define

$$z_1 := \frac{1}{\lfloor d/2 \rfloor} \sum_{i=1}^{\lfloor d/2 \rfloor} v_i \qquad \text{and} \qquad z_2 := \frac{1}{d - \lfloor d/2 \rfloor} \sum_{i=\lceil d/2 \rceil}^{d} v_i$$

to extend some two-dimensional functions to multiple dimensions.

**Styblinski-Tang**

$$\frac{1}{2}\sum_{i=1}^{d}(v_i^4 - 16v_i^2 + 5v_i) \tag{31}$$

**Holder table**

$$10\left|\sin\left(z_1\right)\cos\left(z_2\right)\exp\left(\left|1 - \frac{\|v\|}{\pi}\right|\right)\right| \tag{32}$$

**Flowers**

$$\sum_{i=1}^{d}\left(v_i + 2*\sin\left(|v_i|^{1.2}\right)\right) \tag{33}$$

**Oakley-Ohagan**

$$5\sum_{i=1}^{d}(\sin(v_i) + \cos(v_i) + v_i^2 + v_i) \tag{34}$$

**Watershed**

$$\frac{1}{10}\sum_{i=1}^{d-1}\left(v_i + v_i^2(v_{i+1} + 4)\right) \tag{35}$$

**Ishigami**

$$\sin(z_1) + 7\sin(z_2)^2 + \frac{1}{10}\left(\frac{z_1 + z_2}{2}\right)^4\sin(z_1) \tag{36}$$

**Friedman**

$$\frac{1}{100}\left(10\sin\left(2\pi(z_1 - 7)(z_2 - 7)\right) + 20\left(2(z_1 - 7)\sin(z_2 - 7) - \frac{1}{2}\right)^2\right.$$
$$\left. + 10\left(2(z_1 - 7)\cos(z_2 - 7) - 1\right)^2 + \frac{1}{10}(z_2 - 7)\sin(2(z_1 - 7))\right) \tag{37}$$

**Sphere**

$$-10\|x\|^2 \tag{38}$$

**Bohachevsky**

$$10\left(z_1^2 + 2z_2^2 - 0.3\cos(3\pi z_1) - 0.4\cos(4\pi z_2)\right) \tag{39}$$

**Wavy plateau**

$$\sum_{i=1}^{d}\left(\cos(\pi v_i) + \frac{1}{2}v_i^4 - 3v_i^2 + 1\right) \tag{40}$$

**Zig-zag ridge**

$$\sum_{i=1}^{d-1}\left(|v_i - v_{i+1}|^2 + \cos(v_i)(v_i + v_{i+1}) + v_i^2 v_{i+1}\right) \tag{41}$$

**Double exponential**   with $\sigma = 20, m = 3$, and $\mathbf{1}$ being the $d$-dimensional vector of ones,

$$200\exp\left(-\frac{\|v - m\mathbf{1}\|^2}{\sigma}\right) + \exp\left(-\frac{\|v + m\mathbf{1}\|}{\sigma}\right) \tag{42}$$

**ReLU**   dimension: any

$$-50 \sum_{i=1}^{d} \max(0, v_i) \tag{43}$$

**Rotational**

$$10\mathrm{ReLU}(\arctan2(z_2 + 5, z_1 + 5) + \pi) \tag{44}$$

**Flat**   dimension: any

$$0 \tag{45}$$

## G   Failure modes

### G.1   On the observability of the energy functionals

When multiple energy functionals are at play or are being considered to describe the evolution of the particles, it is not always possible to recover the true description of the energies. That is, the energy functionals are, in general, not observable. To exemplify this, consider particles initially distributed according to a Gaussian distribution $\mu(t_0) = \mathcal{N}(0, 1)$ with zero mean and unit variance, diffusing as

$$\mathrm{d}X(t) = \alpha X(t)\mathrm{d}t + \sqrt{2\beta}\mathrm{d}W(t),$$

i.e., minimizing the potential energy $V(x) = -\frac{\alpha}{2}x^2$ and the internal energy. When $\alpha = 0$, at any time $t$ the particles distribution is $\mu_\beta(t) = \mathcal{N}(0, 1 + 2\beta(t - t_0))$. When $\beta = 0$, instead, each particle position at any time $t$ is $x(t) = e^{\alpha t}x(0)$ and, thus, the particles distribution is $\mu_\alpha(t) = \mathcal{N}(0, e^{2\alpha t})$. By super-position, the particles distribution at each time $t$ is $\mu(t) = \mathcal{N}(0, e^{2\alpha t} + 2\beta t)$.

Suppose we only get two snapshots, $\mathcal{N}(0, 1)$ and $\mathcal{N}(0, \sigma_1^2)$ of the population, at time $t = 0$ and at time $t = T_1$. For an external observer, there are infinitely many solutions describing the diffusion process: one for each pair $(\alpha, \beta)$ that satisfies

$$e^{2\alpha T_1} + 2\beta T_1 = \sigma_1^2.$$

While this corner case is easily resolved with a third observation $\mathcal{N}(0, \sigma_2^2)$ at time $T_2$, it showcases that it is not obvious or necessarily easy to learn the underlying energies. Even when multiple empirical observations are provided, JKOnet$^*$ may fail to distinguish the different energy components.

It should be noted, however, that JKOnet$^*$ is, to the best of our knowledge, the only method currently capable of recovering the distinct energy components. Other methods would associate all the effects to the potential energy, failing to realize there the internal energy is also at play.

In these corner cases, our method can still provide a proxy for prediction, but limited insights. We thus believe that understanding under which conditions there is a unique selection of energy functionals that describe the diffusion process is a fundamental theoretical question.

### G.2   When the process is not a diffusion

While JKOnet$^*$ expands upon the state-of-the-art in terms of expressive capabilities, it is still bound to diffusion processes described by (14). To see what happens when this is not the case but one still deploys JKOnet$^*$, suppose we are given observations of particles $x(t) = \begin{bmatrix} x_1(t) & x_2(t) \end{bmatrix}^\top$ evolving according to

$$\dot{x}(t) = f(x(t)) = \begin{bmatrix} x_2(t) - x_1(t) \\ x_2(t) \end{bmatrix}. \tag{46}$$

There is no interaction between the particles and no noise. Moreover, there is no function $V$ such that $f(x) = -\nabla V(x)$. We sample 1000 particles in $[-4, 4]^2$ and observe their evolution over $T = 5$ steps, with $\tau = 0.01$. We train JKOnet$^*$ with the inductive biases $\theta_2 = \theta_3 = 0$ for 200 epochs and juxtapose in Figure 11 the resulting predictions with the actual particles trajectory, and the potential vector field with the actual vector field. Not surprisingly, the results are not satisfying. In view of Helmholtz decomposition theorem, we can write $f(x) = \nabla V(x) + \nabla \times \psi(x)$, for some $\psi(x)$. Thus, a promising direction for future work on the topic is the design of methods capable also of learning $\psi(x)$.

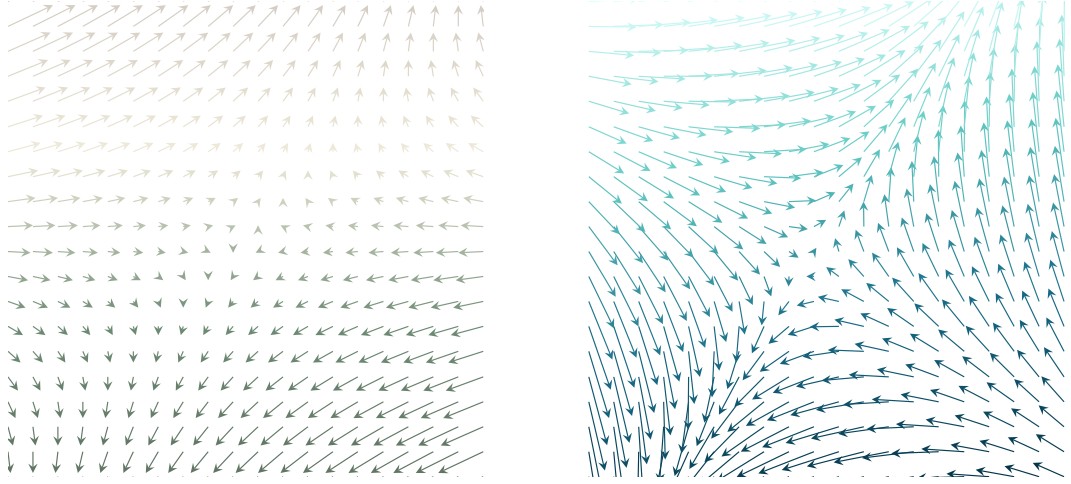

Figure 11: Vector field real (green) and estimated with JKOnet$^*$ (blue) of the process in (46), which contains a solenoidal component. Not surprisingly, when the underlying process is not in the form of (14) the model fails to recover the correct energy terms.

