# OpenReview forum: "Learning diffusion at lightspeed"
_NeurIPS.cc/2024/Conference — NeurIPS 2024 oral_

### Official Review · Reviewer_vFxe · 2024-06-30

**Soundness:** 4
**Presentation:** 3
**Contribution:** 4
**Rating:** 7
**Confidence:** 3

**Summary:**

This paper considers learning diffusion dynamics from observational data of populations over time, identified as learning the energy functional in Equation 3.  Past research has confronted this inverse problem via complex bilevel optimization, limited to potential energies.  This paper proposes an alternative model JKOnet* that can work with potential, internal, and interaction energies, efficiently minimizes a quadratic loss instead of a complex bilevel optimization, has much lower computational complexity, and out-performs baselines in simulations.  A variant for linearly parameterized functionals has a closed form solution.  The paper's new method reconsiders the JKO scheme using first-order optimality conditions, resulting in decompose the problem into first computing optimal transport plans between adjacent populations and then optimizing a loss for fixed plans.

**Strengths:**

- Inferring diffusion dynamics from observational data is a difficult and significant problem for which this paper appears to provide a solid contribution.  The paper substantially improves upon JKOnet in terms of multiple directions: better performance (Figure 3), simpler optimization objective (Equation 11), better scalability and efficiency (e.g. Table 1, Section 4.2), and improved generality (Table 1, Section 4.3).  These dimensions are analyzed in experiments across a range of different energy functionals, where the gains are shown in log-scale displaying orders of magnitude improvement.  The paper makes a convincing argument for using JKOnet* over JKOnet.
- The methodology appears quite strong, well-motivated, and original, with solid intuition given by the authors throughout the paper.

**Weaknesses:**

Minor weaknesses:
- While the results are strong, occasionally the language feels too imprecise.  For example, "runs at lightspeed" seems inaccurate compared to "runs very efficiently".  The authors also mention that they rely upon weeks-old advancements in optimization in the abstract which seems unneeded.
-  The paper is generally very well-written except for the introduction which could use editing.  It introduces a lot of terminology and details from past research.  Similarly, Figure 1 is referenced multiple times including in the introduction but it was hard to understand until after reading Section 3.
- The construction of the optimal transport plans does not seem to be included in the computational complexity comparisons.  While this is computed once for JKOnet*, it is additional expense over JKOnet.

**Questions:**

1.  What is JKOnet_l in Table 1?

2.  In Section 4.2, the authors conclude that JKOnet* is well-suited for high-dimensional tasks.  Does this include computing the optimal transport maps?

3.  The discussion in Figure 3 in the text focuses primarily on the speed improvement, yet the performance gains are also quite large, including seemingly between JKOnet* and JKOnet*_l.  Can the authors comment on why the linear parameterization was useful in their experiments?

**Limitations:**

Limitations are adequately addressed in Section 5

---

> ### Author Rebuttal · Authors · 2024-08-04
>
> ### Weakness 1
> We appreciate the suggestions to improve the exposition and agree with the reviewer's comment. Specifically, we now removed "phenomenal", "runs at lightspeed" and "few weeks old advancements" from the abstract and introduction, rephrasing so to be more factual in the exposition, focusing on the contributions. Thanks!
>
> ### Weakness 2
> We agree that the introduction blends with the related works rather abruptly. We thus prepared a revised version in which we smoothed it, gradually adding information for the reader less familiar with the literature.
>
> For instance, we will add the paragraph "However, the JKO scheme entails an optimization problem in the probability space. Thus,  the problem of finding the energy functional that minimizes a prediction error (w.r.t. observational data) takes the form of a computationally-challenging infinite-dimensional bilevel optimization problem, whereby the upper-level problem is the minimization of the prediction error and the lower-level problem is the JKO scheme." to better introduce the reader to what the bilevel optimization problem is about.
>
> We also removed the focus from Figure 1 in the introduction.
>
> ### Weakness 3
> We thank the reviewer for pointing out this difference, which in fact is a strength of our method which we did not highlight appropriately and we hope we now did in the Remark 3.6 we prepared for the next revision of the paper. In a nutshell, it is true that $\mathrm{JKOnet^*}$ requires the construction of the optimal transport couplings beforehand. However, $\mathrm{JKOnet}$ constructs a new optimal transport plan at each iteration depending on the current estimate of the potential, whereas $\mathrm{JKOnet^*}$ needs to do so only once, at the beginning. We prepared a revision of the paper in which we added the time required to compute the optimal transport couplings in Section 4.1 ($0.03 \pm 0.01$s), and we provided an analysis of the comparison between Sinkhorn and plain linear programming for our scope in the newly added Figure 10 in Appendix C.2, which we now expanded (we report Figure 10 in the PDF attached to this rebuttal, and we will include the extended version of the Appendix C.2 for the next revision of the paper). We also better argued why the scaling to high-dimensions remains unaffected by the computation of the optimal transport couplings: the dimensionality affects only the construction of the cost matrix in the linear program, and otherwise the computational complexity of the linear program only relates to the number of particles. When the number of particles grows, one can apply the same batching applied in other methods to pre-compute the couplings. We add details regarding these practical considerations in the revised Appendix C.2. Finally, following the comments from reviewer vNRb, we introduced a real-world case study on learning and predicting molecular processes (see Figure 5 and the comparison table in the attached document). This way, we hope we strengthen our contributions not only by achieving state-of-the-art on a real-world application but doing so in under a minute of training (including the computation of the optimal transport couplings) compared to the hours required by the other methods.
>
> ### Question 1
> Thanks for pointing out the notation confusion. We defined $\mathrm{JKOnet^*_l}$ only later on, so now we introduced in the caption to clarify we refer to the linear parametrization.
>
> ### Question 2
> Thanks for pointing this out. We agree that this is an important point and we will discuss it in the revised version of the paper. In particular, the dimensionality affects only the construction of the cost matrix in the linear program associated with the optimal transport problem, and otherwise the complexity is only related to the number of particles. Specifically, the time required to construct the cost matrix scales linearly with the dimension, and in practice it is minimal and dwarfed by the actual solution of the linear program. When the number of particles grows, one can apply the same batching that is applied in other methods to pre-compute the couplings. We add details regarding these practical considerations in the revision of Appendix C.2. Please see also point 3 above.
>
> ### Question 3
> Thanks for observing the performance gains and pointing out this aspect. We prepared a revision of Section 4.1 in which we added a paragraph highlighting also the performance gains and discussing linear vs non-linear. In particular, the linear approximation has optimality guarantees, as long as the features are sufficiently rich. In high dimensions, however, the choice of feature is challenging and, thus, we recommend resorting to non-linear parametrizations.

---

> > ### Comment · Reviewer_vFxe · 2024-08-09
> >
> > I appreciate the detailed response and additional experiments in the rebuttal, and continue to recommend paper acceptance.

---

### Official Review · Reviewer_fc3q · 2024-07-10

**Soundness:** 4
**Presentation:** 4
**Contribution:** 4
**Rating:** 8
**Confidence:** 4

**Summary:**

The authors study diffusion processes from the perspective of Wasserstein gradient flows. Based on the recent fixed-point characterisation for Wasserstein proximal operator methods, they introduce Jordan-Kinderlehrer-Otto (JKO) type methods for learning potential and interaction energies that govern the diffusion process. Such methods are assuming that a sample of the population distribution at each time step is at hand (not necessearily obtained by tracking individual particles) implying important applications across various fields. While theoretical novelties are present (w.r.t. paper [26] that lies in the foundation of this work), the main contribution is the overall methodology for learning diffusion processes.

**Strengths:**

Paper is, besides minor issues reported bellow, excellently written - very clear, precise and intuitive with well balances technical details between main text and the appendix. Existing ideas are neatly combined to obtain significant improvements of the JKO-type methods and extensive empirical evaluation is presented. The proofs seem correct and well-written.

**Weaknesses:**

While I do not find important weaknesses, I feel that next several small issues can be addressed to further improve readability:

1. When addressing content presented in the appendix it would be good to refer to the section, e.g. see Figure 6 in Appendix A.

2. It would be good to say what $\rho_t$ is in Example 2.1

3. While Table 3.1 reports per-epoch complexity for all the methods, it would be important to note that JKOnet$^*$ have additional computational complexity for solving $T$ OT problems of size $N$ in $d$-dimensions. Detailed remark on the initial computational complexity, depending of the algorithm used, should be reported.

4. In Section 4 it would be helpful to introduce the problems, that is to better explain the task of each experiment and the role of functionals ($V(x)$ ?!)  appearing in Appendix F.  Maybe giving an example on Styblinski-Tang functional appearing in Figures 2, 3 and 4, and then referring to other ones by their names and/or reference equations.

**Questions:**

1. In the implementation of the method, a priori computed optimal transport plans are obtained by solving entropy-regularised OT via Sinkhorn-type algorithms or some other methods?

2. What do you think about the applications and/or limitations of the JKOnet$^*$ for the setting of long-trajectories to infer the behaviour in equilibrium, e.g. detection of meta-stable states of Langevin dynamics?

**Limitations:**

Limitations are addressed adequately.

---

> ### Author Rebuttal · Authors · 2024-08-04
>
> ### Weakness 1
>
> We thank the reviewer for the suggestion. We prepared a revision of the paper in which we added a reference to the appendix when referencing to content related to the appendix.
>
> ### Weakness 2
> Good catch, thank you! We prepared a revision of the paper in which we added the definition of $\rho$ below the Fokker-Plank equation.
>
> ### Weakness 3
> We thank the reviewer for pointing out this difference, which in fact we believe to be a strength of our method which we did not highlight appropriately. To this end,  we included a dedicated remark, Remark 3.6, for the next revision of the paper. In a nutshell, it is true that $\mathrm{JKOnet^*}$ requires the construction of the optimal transport couplings beforehand. However, $\mathrm{JKOnet}$ constructs a new optimal transport plan at each iteration depending on the current estimate of the potential, whereas $\mathrm{JKOnet^*}$ needs to do so only once, at the beginning. We prepared a revision of the paper in which we added the time required to compute the optimal transport couplings in Section 4.1 ($0.03 \pm 0.01$s), and we provided an analysis of the comparison between Sinkhorn and plain linear programming for our scope in the newly added Figure 10 in Appendix C.2, which we now expanded (we report Figure 10 in the PDF attached to this rebuttal, and we will include the extended version of the Appendix C.2 for the next revision of the paper). We also better argued why the scaling to high dimensions remains unaffected by the computation of the optimal transport couplings: the dimensionality affects only the construction of the cost matrix in the linear program, and otherwise the computational complexity of the linear program only relates to the number of particles. When the number of particles grows, one can apply the same batching applied in other methods to pre-compute the couplings. We add details regarding these practical considerations in the revised Appendix C.2. Finally, following the comments from reviewer vNRb, we introduced a real-world case study on learning and predicting molecular processes (see Figure 5 and the comparison table in the attached document). This way, we hope we strengthen our contributions not only by achieving state-of-the-art on a real-world application but doing so in under a minute of training (including the computation of the optimal transport couplings) compared to the hours required by the other methods.
>
> ### Weakness 4
>
> Good point, thank you for the comment. We have a dedicated section in Appendix B, which we now expanded to discuss the prediction schemes as well, and we have added an introduction to the problems in the experimental section. In particular, we added the data-generation equation, in which the role of the functionals $V(x)$ is apparent: $x_{t+1} = x_t - \tau \nabla V(x_t)$. We also added the name and equation of each functional in the figures where it was not listed (e.g., Figure 2).
>
> ### Question 1
> Thanks for the question. In our implementation, we solved the optimal transport problems via plain linear programming (using the POT python library). We prepared an updated version of Appendix C.2 that we will include in the revision of the paper that contains an analysis of the impact of different ways of solution algorithms on the final outcome in terms of computational time during pre-processing and Wasserstein error (see Figure 10 in the attached PDF). In particular, we conclude that, as long as the couplings are close to the correct one, the algorithm used to compute them does not impact the performance of $\mathrm{JKOnet^*}$. Since small regularizers slow down the Sinkhorn algorithm, we opted to directly solve the linear program (without regularization). In general, the solver choice can be considered an additional knob that researchers and practitioners can tune when deploying $\mathrm{JKOnet^*}$. Please see also point 3 above.
>
> ### Question 2
> Once an energy functional through $\mathrm{JKOnet^*}$ is learned, the equilibrium state of the system can be inferred in two ways. A simple approach consists of running sufficiently many iterations of the JKO scheme until an equilibrium is reached. Alternatively, the equilibrium state is well-known to be the minimum of the energy functional. Thus, the equilibrium state can be inferred by computing the probability distribution which minimizes the energy functionals, using tools from optimization in the probability space.

---

> > ### Comment · Reviewer_fc3q · 2024-08-08
> > **Acknowledgement of the rebuttal**
> >
> > I thank the authors for their rebuttal. I remain confident of the quality of their paper, suggest the acceptance and keep my score.

---

### Official Review · Reviewer_vNRb · 2024-07-10

**Soundness:** 3
**Presentation:** 3
**Contribution:** 3
**Rating:** 6
**Confidence:** 3

**Summary:**

This paper introduces JKOnet*, a new method for learning diffusion processes from data. It uses first-order optimality conditions of the JKO scheme instead of complex bilevel optimization. JKOnet* can recover potential, interaction, and internal energy components of diffusion processes. The authors provide theoretical analysis and experiments showing JKOnet* outperforms baselines in accuracy, speed, and ability to handle high-dimensional data. They also derive a closed-form solution for linearly parameterized functionals. JKOnet* offers improved computational efficiency and representational capacity compared to existing approaches for modeling diffusion dynamics from population data.

**Strengths:**

- Develops JKOnet*, a method using first-order optimality conditions of the JKO scheme to learn diffusion processes, avoiding bilevel optimization and improving computational efficiency.
- Provides theoretical analysis and proofs for JKOnet*, including a closed-form solution for linearly parameterized functionals, backed by comprehensive experiments across various test functions.
- Demonstrates improved performance in terms of Wasserstein error and computation time compared to existing methods like JKOnet, especially in high-dimensional settings.
- Enables recovery of potential, interaction, and internal energy components of diffusion processes, expanding the model's applicability to more complex systems and improving interpretability.

**Weaknesses:**

- The experimental evaluation is limited to synthetic datasets. Real-world data applications would strengthen the practical relevance of the method.
- While the paper discusses limitations, it does not thoroughly explore potential failure cases or boundary conditions where JKOnet* might underperform.
- The paper does not provide a comprehensive comparison with other recent approaches in learning diffusion processes beyond JKOnet, which could provide broader context for the method's improvements.

**Questions:**

- The authors demonstrate JKOnet*'s performance on synthetic datasets. How well does the method perform on real-world diffusion processes? Additional evaluations on empirical data would help understand the method's practical applicability.
- The paper focuses comparison mainly with JKOnet. How does JKOnet* compare to other recent approaches in learning diffusion processes?
- In Section 3.4, the authors discuss different parameterizations. How sensitive is JKOnet* to the choice of neural network architecture for the non-linear parameterization case?

**Limitations:**

The author discusses limitations in section 5

---

> ### Author Rebuttal · Authors · 2024-08-04
>
> ### Weakness 1
>
> We thank the reviewer for suggesting us one way to strengthen the presentation of our contributions.
>
> We deployed our method to learn the diffusion dynamics of embryoid body single-cell RNA sequencing (scRNA-seq) data [1], a popular benchmark in the literature, and compared our results with nine other recent methods in the literature. We discuss the application and the results in the newly added Section 4.4, which will appear in future versions of the paper, and we report the related figure and table in the PDF attached to this response. We briefly summarize our experimental setting and results below.
>
> Experimental setting (briefly): We follow the same data pre-processing as in [2,3]; in particular, we use the same processed artifacts of the embryoid data provided in their work, which contains the first 100 components of the principal components analysis (PCA) of the data and, following [2,3], we focus on the first five. We train on $60\%$ of the data at each time-step and test $\mathrm{JKOnet^*}$ to predict the evolution of the left-out data.
>
> Results: $\mathrm{JKOnet^*}$ outperforms all existing methods in the literature (see the results table in the PDF attached). Importantly, our training takes less than a minute (including the computation of optimal transport plans), whereas the training time of all other existing methods takes hours.
>
> We visualize the 2 principal components and the interpolations obtained with our method in Figure 5. Note that, for this application, we used a time-varying potential energy, of which we plot the level curves.
>
> ### Weakness 2
>
> We believe our approach might underperform in the following two cases.
>
> First, while diffusion processes include many real-world phenomena, in some real-world applications it might be unknown if the particles are undergoing diffusion. If this is not the case, $\mathrm{JKOnet^*}$ might underperform. For instance, in the absence of noise and interaction energy, if the vector field is not the gradient of some potential energy function (e.g., it includes a solenoidal component $\nabla \times \psi$ for some function $\psi: \mathbb{R}^d \to \mathbb{R}^d$), we cannot expect $\mathrm{JKOnet^*}$ (as well as any other method learning a potential) to infer a reasonable potential. We prepared a dedicated section that we will include in the revision of the paper to discuss this failure mode.
>
> Second, when learning both potential and interaction energy and noise level, there might be observability issues that prevent the distinction of the different components of the energy functional (e.g., a discrete-time population-level effect might be explained both by a potential energy and a noise level).
>
> While we did not experience this issue in our experiments in Section 4.3, we are not aware of rigorous guarantees that ensure observability. In a dedicated section in Appendix G, we provide a small-scale analytical example illustrating this issue. As $\mathrm{JKOnet^*}$ is the only method capable of simultaneously learning all three energy components, we believe it can serve as the baseline to investigate this observability issue.
>
> ### Weakness 3
>
> We compared our method with others in our real-world application, discussed in 1) above and in Section 4.4 in the revised version of the paper.
>
> ### Question 1, 2
>
> Please refer to our answer to Weakness 1) above.
>
> ### Question 3
>
> In our experiments, we considered only vanilla parametrizations: two-layers MLP with 64 neurons in each layer, to compare directly with the other works in the literature. What is certainly required is a network that is expressive enough to approximate the energy functional of interest, so standard rationales apply for the choice of activation functions (we use softplus), dimension of the networks, etc. One of the limitations of our works is the data domain (we do not explore, e.g., images). We plan in future work to do so, and in that case more care will be needed to determine the most suitable architecture. Given that the same architecture and hyperparameters worked well across all the experiments (including the real-world experiment), we are confident to state that the learning algorithm itself is not particularly sensitive to the network architecture, which needs of course be chosen to be suitable to represent the energy in the application of choice. We also believe that exciting future work can be done in terms of understanding how different architectures can capture potential, internal, and interaction energies more efficiently: can e.g., transformers be used to learn an interaction energy more efficiently than a vanilla MLP?
>
> [1] "Visualizing structure and transitions in high-dimensional biological data" by Kevin R Moon, David van Dijk, Zheng Wang, Scott Gigante, Daniel B Burkhardt, William S Chen, Kristina Yim, Antonia van den Elzen, Matthew J Hirn, Ronald R Coifman, et al. (2019)
>
> [2] "Improving and generalizing flow-based generative models with minibatch optimal transport" by Alexander Tong, Nikolay Malkin, Guillaume Huguet, Yanlei Zhang, Jarrid Rector-Brooks,
> Kilian Fatras, Guy Wolf, and Yoshua Bengio. (2023)
>
> [3] "Deep momentum multi-marginal schr\"{o}dinger bridge" by Tianrong Chen, Guan-Horng Liu, Molei Tao, and Evangelos A Theodorou. (2023)

---

> > ### Comment · Reviewer_vNRb · 2024-08-10
> >
> > Thank you for your response. These addressed my questions. I have raised my score.

---

### Official Review · Reviewer_CXdg · 2024-07-18

**Soundness:** 3
**Presentation:** 3
**Contribution:** 2
**Rating:** 5
**Confidence:** 3

**Summary:**

This paper studies the problem of learning a diffusion process from samples. It proposes a new scheme based on learning the "causes mismatch" of the process, rather than the "effects mismatch" as in previous works. The new method is significantly more efficient than the schemes from prior works, and works well in practice.

**Strengths:**

The paper is well-written, and the scheme proposed seems to work well in practice on the examples it was tested on. The loss is intuitive, and resembles the score-matching loss from diffusion models, but is the analogous version for arbitrary diffusion processes. Overall, this seems like a paper that people at NeurIPS would be interested in.

**Weaknesses:**

I am not familiar enough with the literature, but it seems surprising to me that this scheme has never been proposed before. In particular, the loss is exactly the score-matching in the case of diffusion models, and there are works [1], [2] that have proposed a similar loss for arbitrary diffusion processes.

[1]: https://arxiv.org/abs/2208.09392
[2]: https://arxiv.org/abs/2209.05442

**Questions:**

1) Can you provide a more thorough comparison with prior literature, especially the works I have linked above?

---

> ### Author Rebuttal · Authors · 2024-08-04
>
> We believe the problem of score-matching in diffusion models to be fundamentally different from the one in our paper.
> In score-matching, one tries to "reverse" the time of a known diffusion process, e.g., to recover the uncorrupted state of a corrupted image.
> In our setting, instead, we use observational population data to learn the energy functional underlying an unknown diffusion process. Our goal is not to "reverse" the time of a given diffusion process to reconstruct its initial condition but rather to learn an unknown diffusion process to perform forward-in-time predictions.
>
> From a technical perspective, our methodology heavily relies on optimal transport theory and tools from optimization in the probability space. Indeed, our loss can be interpreted as the "error" in satisfying a first-order optimality condition in the Wasserstein space.
> To the best of our knowledge, this approach has not previously appeared in the literature.
> For instance, the loss function in reference [1] suggested by the reviewer is constructed using tools from stochastic differential equations: Their loss function minimizes the errors in estimating the term $\nabla_x\log q_t(x)$ which appears when "reversing" the time of a (known) stochastic differential equation. For this reason, their loss cannot be reconciled to ours (in which the stochastic differential equation is unknown).
>
> [1] "Soft diffusion: Score matching for general corruptions" by Daras, G., Delbracio, M., Talebi, H., Dimakis, A. G., \& Milanfar, P. (2022).

---

### Author Rebuttal · Authors · 2024-08-04

We thank the reviewer for their time and constructive feedback. Our main changes can be summarized as follows:

First, we applied our methodology to real data in single-cell diffusion dynamics and compared our results with nine existing methods, as requested by reviewer vNRb. In short, our model, $\mathrm{JKOnet^*}$, outperforms all existing methods both in terms of solution accuracy and training time. Remarkably, our training takes less than a minute, in contrast to the hours of all other methods in the literature.

Second, we detailed the computation of optimal transport plans/maps, as requested by reviewers fc3q and vFxe. In a nutshell, $\mathrm{JKOnet^*}$ requires computing optimal transport plans only once, before the training. Conversely, $\mathrm{JKOnet}$ demands re-computing optimal transport plans at each training step. Additionally, we clarified that our numerical experiments rely on the linear programming formulation of optimal transport and included an ablation study to compare the linear programming formulation and Sinkhorn algorithm.

Third, we included a discussion of the failure modes, as requested by vNRb. In short, we envision $\mathrm{JKOnet^*}$ to underperform when the underlying process is not a diffusion process and when observability issues arise (which make the different energy components indistinguishable).
We included a discussion and examples to illustrate these two corner case phenomena (which, however, we did not experience in our experiments).

For a detailed response to each reviewer's question and concern, please refer to the responses below.

We believe that these changes both strengthen our contribution and improve the presentation of our results. We thank again the reviewers for the comments that helped us do so!

---

### Decision · Program_Chairs · 2024-09-25

**Decision:**

Accept (oral)

**Comment:**

The paper presents improved methods for learning energy functionals governing a diffusion process from population snapshots observed over time. Existing work uses the JKO scheme, which provides a variational formulation of the probability path, and uses bilevel optimization to learn the energy functionals based on the mismatch between the predicted probability path and the sequence of observed distributions. In this paper, the authors instead use a recent characterization of first-order optimality conditions of the JKO scheme, and find energy functionals to directly satisfy these conditions, leading to significant computational advantages. Reviewers thought the paper was very strong: “excellently written”, “precise”, “intuitive”, addresses a “difficult and significant problem”, “substantially improves upon JKOnet in terms of multiple directions” and so on. Most of the weaknesses were labeled as minor, and were well addressed by the rebuttal. The meta-reviewer appreciates the author’s promise to tone down some language in the introduction (“phenomenal”, “lightspeed”, “few-weeks-old”) and the addition of experiments on a real benchmark dataset with comparisons to more methods during the rebuttal. These will both strengthen the paper.